# 3D-printable phosphorescent woody materials

Zhijun Chen [1,2], Kai Wang[1,2], Yingxiang Zhai[1,2] ✉, Yujie Bai[1,2], Zijing Pan[1,2], Jingyi Zhou[1,2], Min Wang[1,2], Luyao Wang[1,2], Xue Liu[1,2], Chenhui Yang [1,2], Shouxin Liu[1], Jian Li[1], Chuanling Si [3] ✉, Shujun Li [1,2] ✉, Yiqiang Wu[4] ✉ & Tony D. James [5,6] ✉

The preparation of sustainable biophosphors exhibiting room-temperature phosphorescence (RTP) for additive manufacturing presents both significant scientific promise and substantial synthetic challenges. To address this technological gap, with this research, we engineer CX-Wood using rational molecular design by grafting carboxyl-functional groups onto native lignocellulosic matrices, enabling direct ink writing (DIW) using our RTP wood composite. Structural characterization reveals that carboxylation induces (i) partial crystal lattice distortion in the cellulose microfibrils and (ii) enhances the hydrogen-bonding network density, collectively establishing a rigid supramolecular architecture conducive to triplet-state stabilization. This structural modification improves room-temperature phosphorescent performance. Crucially, the introduced carboxyl moieties simultaneously optimize the rheological behavior to yield an aqueous-based phosphorescent ink with exceptional print fidelity. Leveraging this dual functionality, we prepare architecturally complex 3D phosphorescent constructs exhibiting afterglow emission. This biomass-derived platform establishes a green model for manufacturing smart luminescent materials with tailored properties.

Organic room temperature phosphorescent (RTP) materials have received significant attention due to their intrinsic structural flexibility and tunable optical performance[1–3]. As such, organic RTP materials exhibit great potential for bioimaging[4–6], optoelectronics[7–9], anti-counterfeiting applications[10–12] and sensors[13–15]. There are two crucial points for designing organic RTP materials[16–18]: (a) The spin-orbit coupling (SOC) value of the organic chromophore should be promoted so that the intersystem crossing process of the chromophore is strengthened and more triplet excitons are generated for RTP emission. Generally, the SOC value of the chromophore can be increased by decorating the chromophore with heavy atoms or units with lone pairs

of electrons; b) Radiative migration of the as-generated triplet excitons should be facilitated, which can be realized by rigidifying the chromophore.

Recently, producing organic RTP materials from biomass resources has received particular attention considering the sustainability, abundance, and low cost of these resources[19–22]. To date, several biomass resources, such as, wood[23–26], cellulose[27–29], lignin[30–32], hemicellulose[33–35] and natural phenolics[36], have been converted into sustainable RTP materials. Using their various physicochemical properties, these sustainable RTP materials have been further processed into different structures and architectures via solvent casting,

[1]State Key Laboratory of Woody Oil Resources Utilization, Northeast Forestry University, Harbin, China. [2]Key Laboratory of Bio-based Material Science and Technology of Ministry of Education, Northeast Forestry University, Harbin, China. [3]Tianjin Key Laboratory of Pulp and Paper, Tianjin University of Science and Technology, Tianjin, China. [4]State Key Laboratory of Woody Oil Resources Utilization, Central South University of Forestry and Technology, Changsha, Hunan, China. [5]Department of Chemistry, University of Bath, Bath BA2 7AY, UK. [6]School of Chemistry and Chemical Engineering, Henan Normal University, Xinxiang, China. ✉e-mail: zyx1105@nefu.edu.cn; sichli@tust.edu.cn; lishujun@nefu.edu.cn; wuyq0506@126.com; T.D.James@bath.ac.uk

thermoforming or photocuring methods for practical applications[32,37–39]. The shape of RTP materials is just as important as the intrinsic properties, yet most currently used methods are unable to produce complex or customized geometries in a convenient manner[19]. Moreover, most molding methods require the assistance of an external template, increasing the complexity of the material processing.

To conquer these challenges, several RTP formulations suitable for additive manufacturing have already been developed[24,40–42]. Additive manufacturing (AM), also known as three-dimensional (3D) printing, is the latest manufacturing technology with high material efficiency and design flexibility[43,44]. Among the various AM techniques, direct ink writing (DIW) has emerged as the most versatile 3D printing technique for the broadest range of materials[45,46]. DIW allows printing of practically any material, including metals, polymers, concretes, and biomaterials, as long as the precursor ink can be engineered to exhibit appropriate rheological behavior[40,47–51]. This technique acts as an effective pathway to introduce design freedom, multifunctionality, and stability simultaneously into the printed structures.

Nevertheless, most of the as-developed formulations for AM, particularly for DIW printing, require complicated synthetic protocols and contain petrol-derived components. Fully bio-based and easily obtained RTP materials for AM have rarely been reported.

On the other hand, producing RTP materials from natural wood represents a sustainable trend considering the abundance, low cost, renewable, and inherent RTP emission of wood. In fact, DIW has emerged as the most versatile 3D printing technique for natural wood. Specifically, wood has been engineered to exhibit appropriate rheological behavior for DIW printing by introducing thermoplastic plastics (such as PE, PP, PVC, PLA, and so on)[52–55]. Additionally, DIW inks can also be prepared using wood by extracting components (cellulose, hemicellulose, and lignin) and subsequent modification in order to achieve appropriate rheological properties. For example, Thakur et al. reported on the introduction of nanocellulose into woody components to tune the rheological behavior to create DIW inks[51]. Nevertheless, it is still a big challenge to obtain woody materials with RTP and printable properties at the same time.

Here, we developed **CX-Wood** as printable RTP materials for DIW by decorating wood powders with carboxylic acid moieties (Fig. 1a). The as-obtained CX-Wood exhibited RTP emission with a lifetime of 358.7 ms and was flexibly converted into different shapes using DIW technology (Fig. 1b, c).

## Results

### Characterization and RTP of CX-Wood

CX-Wood was prepared by the carboxymethylation of natural wood. The XPS spectra indicated that the ratio of O atoms was greatly enhanced, confirming a successful reaction (Supplementary Fig. 1). High-resolution C spectra suggested that CX-Wood exhibited a new signal of -O-C=O at 289.5 eV, confirming successful modification (Fig. 2a)[56,57]. FT-IR analysis was also consistent with the results of the XPS spectra. With signals associated with the carbonyl moieties at 1595 cm$^{-1}$ being enhanced after decorating natural wood using carboxymethylation (Supplementary Fig. 2)[58,59]. Notably, the size of wood particles was closely related to the substitution degree of carboxylic acid moieties on the CX-Wood. Generally, wood powders of a reduced size exhibited a higher substitution degree (Supplementary Fig. 3). This was because the reduced size was beneficial for interaction with the liquid solution, thus increasing the reaction efficiency. Besides this, degrees of substitution (DS) of CX-Wood also depend on the reaction time or temperature (Supplementary Table 1 and Supplementary Fig. 4). The optimized DS was obtained under the following conditions: alkali treatment at 35 °C for 60 min, followed by ether-forming reaction at 80 °C for 90 min. To evaluate the environmental performance of CX-Wood, we conducted a comparative life cycle assessment and cost analysis of CX-Wood against other reported materials for wood-based 3D printing. The results demonstrate the exceptional environmental performance of CX-Wood ink, in particular its global warming potential (GWP) was only 27.6% that of simulated wood ink and 56.4% of PALF ink. The GWP analysis confirms a significantly lower carbon footprint for CX-Wood (12.03 kg $CO_2$ eq). Consequently, compared to currently reported wood-based 3D printing materials, CX-Wood ink production proves to be both cost-effective and environmentally sustainable (Supplementary Fig. 5 and Supplementary Table 2).

Subsequently, the optical properties of CX-Wood were evaluated. The CX-Wood exhibited fluorescence emission centered at 460 nm and phosphorescence emission centered at 505 nm (Supplementary Fig. 6). Notably, CX-Wood exhibited more intensive and longer RTP emission than the untreated wood (Fig. 2b, c). Time-resolved

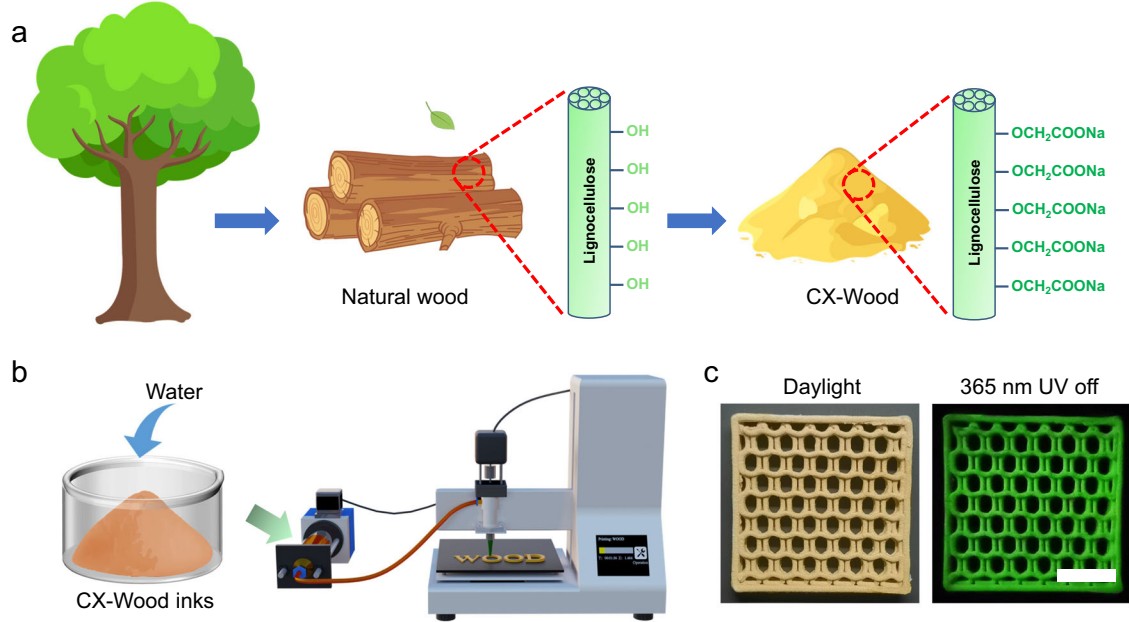

**Fig. 1 | Preparation of CX-Wood. a** Schematic showing the preparation of room temperature phosphorescent CX-Wood from natural wood. **b** Schematic of the CX-Wood printing process. **c** Printed sample of CX-Wood in daylight and with 365 nm UV light off, scale bar = 2 cm.

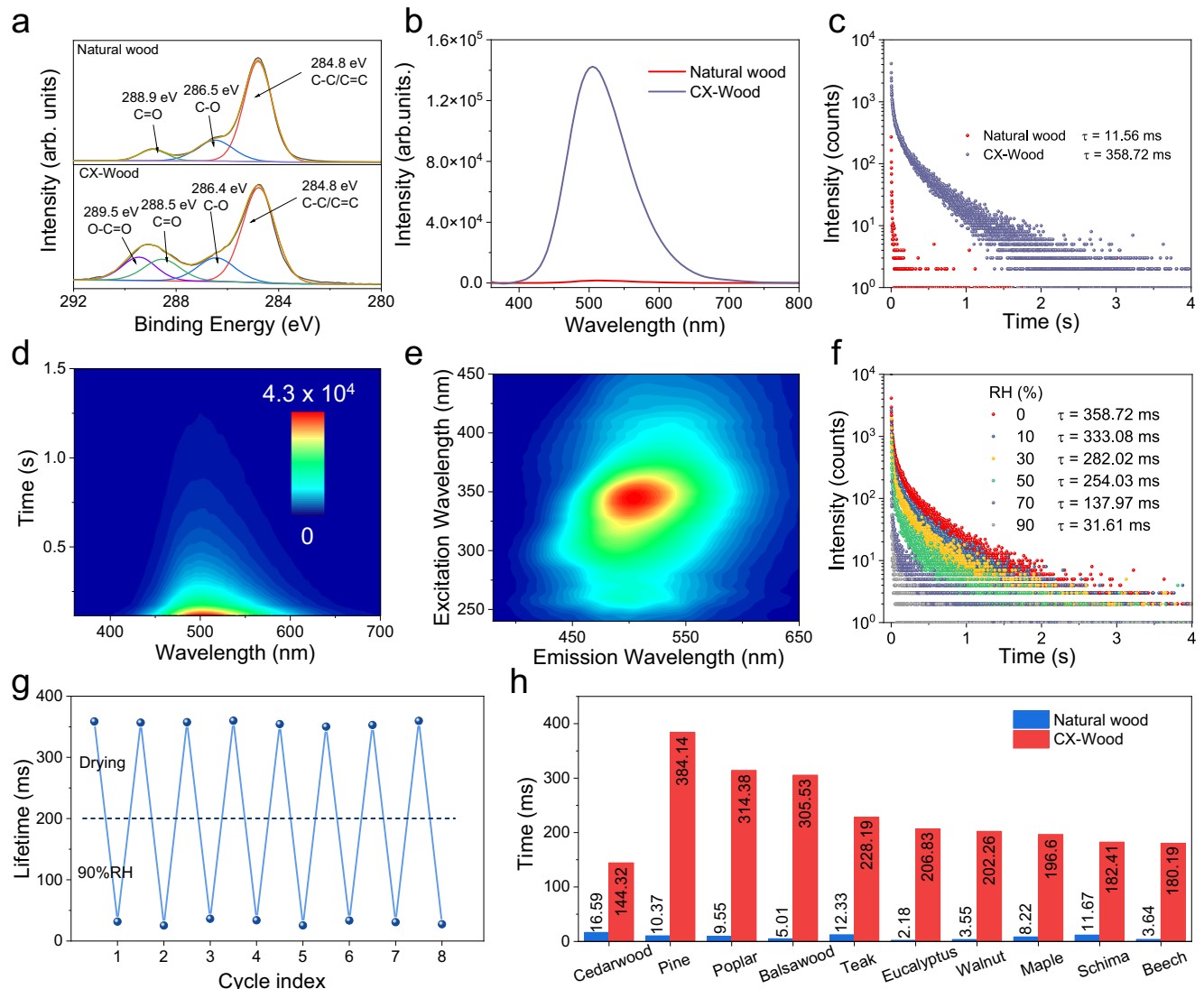

**Fig. 2 | RTP emission of CX-Wood. a** High resolution XPS C1s spectra of natural wood and CX-Wood. **b** Phosphorescence spectra of natural wood and CX-Wood. **c** Phosphorescence decay profiles of natural wood and CX-Wood. **d** Time-dependent phosphorescence emission of CX-Wood. **e** RTP emissions of CX-Wood using different excitation wavelengths. **f** Lifetime decay profiles of CX-Wood determined under different relative humidity (RH) conditions. **g** RTP lifetimes of CX-Wood after "drying-humidity" cycles. **h** RTP lifetime of cedarwood, pine, poplar, balsawood, teak, eucalyptus, walnut, maple, schima, beech and the corresponding CX-Wood. The measurement conditions for the RTP spectra and lifetime corresponded to 10 ms delay, room temperature, excitation wavelength = 340 nm (The data are from a single measurement).

spectroscopy indicated that CX-Wood displays a long-lasting and stable afterglow emission and retained weak phosphorescence emission for as long as 1.2 s (Fig. 2d). In comparison, the commercial carboxymethyl cellulose exhibited inferior phosphorescence intensity and phosphorescence lifetime compared to CX-Wood (Supplementary Fig. 7). Interestingly, both the RTP intensity and lifetime of CX-Wood was increased by enhancing the substitution degree of carboxylic acid moieties on the natural wood (Supplementary Fig. 8). Specifically, the RTP lifetime increased from 35.4 ms to 358.7 ms when the substitution degree increased from 0.066 to 0.5760. The phosphorescence quantum yield of CX-Wood also increased from 0.93% to 4.60% when the substitution degree increased from 0.066 to 0.5760 (Supplementary Fig. 9). Significantly, the formulas used in the literature exhibited shorter lifetimes with values of 28.97 ms[51] and 72.74 ms[60] (Supplementary Table 2).

To evaluate the effect of sodium acetate ions on the RTP performance of CX-Wood, additional Na+ was added to the CX-Wood system.

The measured phosphorescence spectrum and phosphorescence lifetime revealed no significant changes in the phosphorescent behavior of CX-Wood, confirming that ionic interactions do not influence its RTP properties (Supplementary Fig. 10).

More interestingly, CX-Wood exhibited excitation/temperature-dependent RTP emission. The RTP wavelength red shifted from 490 nm to 540 nm when the excitation wavelength was increased from 280 nm to 400 nm (Fig. 2e). While the RTP lifetime decreased from 500.3 ms to 36.1 ms when the temperature increased from 77 K to 400 K (Supplementary Fig. 11). Such temperature-dependent RTP emission is explained by non-radiative migration of triplet excitons being promoted at high temperatures. Additionally, RTP emission of CX-Wood was sensitive to humidity. Exposure of CX-Wood to a humid environment decreased the RTP lifetime and intensity (Fig. 2f and Supplementary Fig. 12). However, the quenched lifetime increased again after drying the CX-Wood. The lifetime did not obviously change when the whole process was recycled several times, indicating the

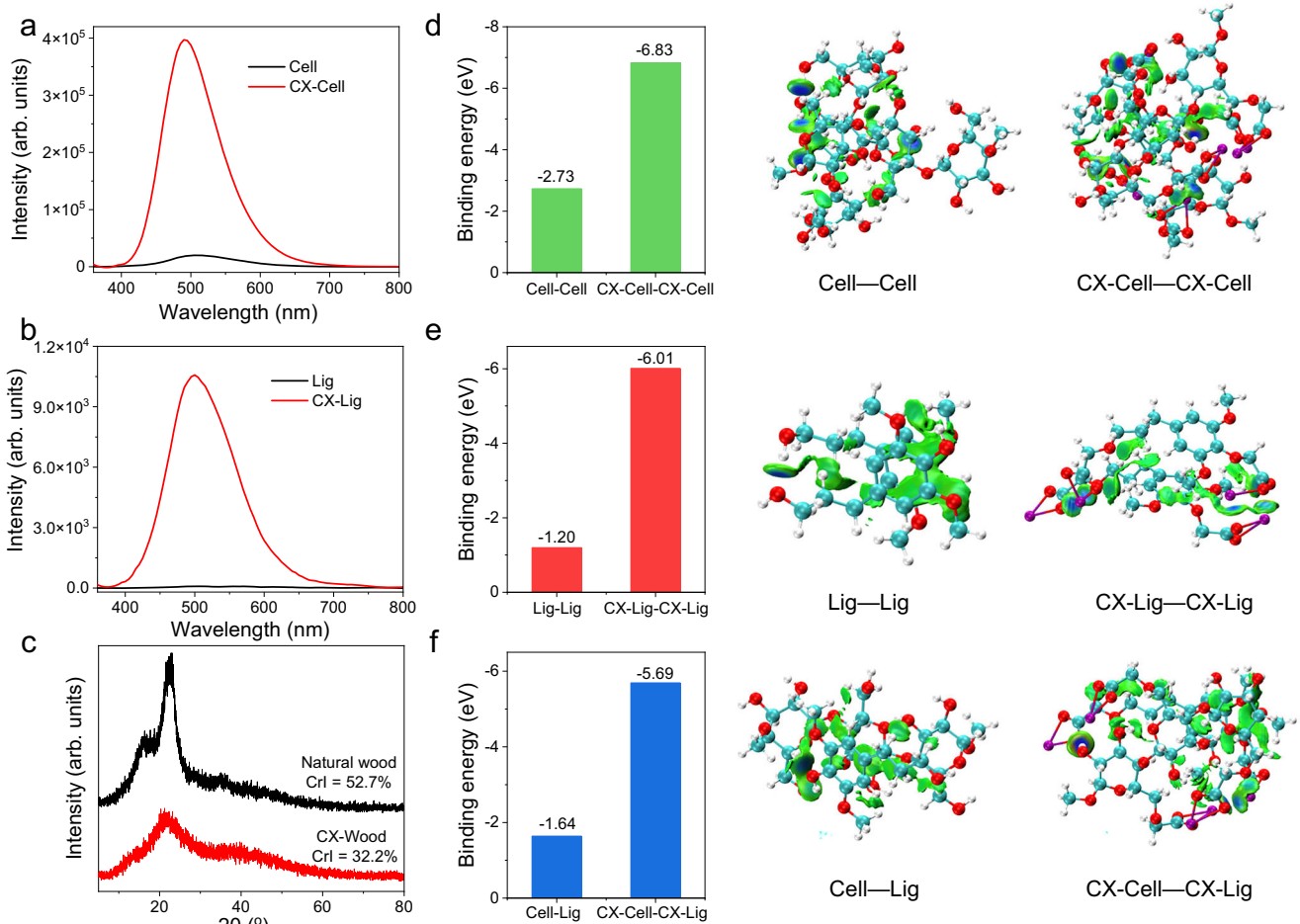

**Fig. 3 | Mechanistic study. a** Phosphorescence spectra of cellulose and CX-Cellulose. **b** Phosphorescence spectra of lignin and CX-Lignin. **c** XRD pattern of natural wood and CX-Wood. **d** Calculated interaction of Cellulose−Cellulose (Cell−Cell) and CX-Cellulose−CX-Cellulose (CX-Cell−CX-Cell). **e** Calculated interaction of Lignin−Lignin (Lig−Lig) and CX-Lignin−CX-Lignin (CX-Lig−CX-Lig). **f** Calculated interaction of Cellulose−Lignin (Cell−Lig) and CX- Cellulose−CX-Lignin (CX-Cell−CX-Lig). The measurement conditions for the RTP spectra corresponded to 10 ms delay, room temperature, excitation wavelength = 340 nm (The data are from a single measurement).

optical robustness of CX-Wood (Fig. 2g). We found that adding borax to the printed materials further increases the rigidity of the system, resulting in an increased RTP lifetime. The lifetime of the modified CX-wood can reach up to 1.06 s, and at the same time, a green afterglow lasting for up to 8 s can be observed with the naked eye (Supplementary Fig. 13). Moreover, RhB can also be introduced into CX-wood. The RTP emission of CX-Wood overlapped with the absorbance of RhB, so that red afterglow emission was observed from CX-wood via a TS-FRET strategy (Supplementary Fig. 14)[61]. The as-obtained CX-Wood/RhB exhibited an RTP emission at 605 nm, which was consistent with its fluorescence spectrum. The lifetime of the red afterglow emission from RhB in the CX-Wood/RhB was 78.08 ms.

To confirm the generality of this method, different types of natural wood were converted into CX-Wood, and all of them exhibited intense and long-lived RTP emission (Fig. 2h and Supplementary Fig. 15).

## Mechanistic study

Natural wood consists of holocellulose (cellulose and hemicellulose) and lignin[62]. In which all the components exhibit RTP emission via either clustering-induced emission[63] or confinement-induced emission[64]. To understand which component contributed to the RTP emission of CX-Wood when compared to natural wood, several control samples, including CX-Cellulose and CX-Lignin, were prepared via

decorating cellulose and lignin using carboxymethylation, correspondingly. CX-Cellulose was used as the representative sample for modified holocellulose in the CX-Wood. CX-Lignin was regarded as the representative sample for modified lignin in the CX-Wood. Interestingly, both cellulose and lignin exhibited enhanced RTP intensity after chemical modification (Fig. 3a, b, and Supplementary Fig. 16). These results indicated that both holocellulose and lignin contributed to the enhanced RTP emission of CX-Wood. Furthermore, CX-Cellulose exhibited an emission in the range of 465−525 nm (Supplementary Fig. 17a). While CX-Lignin exhibited an emission in the range of 475−540 nm (Supplementary Fig. 17b). Thus, the emission peak of CX-Wood in the 480−500 nm range is predominantly attributed to CX-Cellulose, while features in the 500−540 nm range arise primarily from CX-Lignin. Meanwhile, it can be observed that a short-wavelength excitation (250−310 nm) predominantly excites CX-Lignin chromophores, while long-wavelength excitation (310-430 nm) primarily activates CX-Cellulose chromophores (Supplementary Fig. 17). These results further indicate that emission from these components merged into the RTP emission from CX-Wood, and also explain the excitation-dependent RTP emission of CX-Wood.

To further understand such enhancement, the XRD pattern of CX-Wood was analyzed. Interestingly, CX-Wood exhibited decreased crystallinity when compared with natural wood (Fig. 3c). These results indicated that the crystalline region of cellulose in natural wood was

disturbed during the synthetic modification. The modified cellulose chains become amorphous and re-arranged[65]. Therefore, destroying the crystalline domain and creating an amorphous arrangement was beneficial for enhancing hydrogen bonding interactions within cellulose. This result was further confirmed by FT-IR analysis. The signals of the hydroxyl moieties shifted from 3345.5 cm$^{-1}$ to 3315.5 cm$^{-1}$ after introducing carboxylic acid moieties to wood, suggesting intensive hydrogen bonding interactions existed in CX-Wood (Supplementary Fig. 2)[66].

Density functional theory (DFT) calculations indicated that the modified cellulose (CX-Cellulose) formed more intensive hydrogen bonding interactions than untreated cellulose (Fig. 3d, model structure in Supplementary Fig. 18). Additionally, CX-Lignin also exhibited more intense hydrogen bonding interactions than untreated lignin as determined by the DFT calculations (Fig. 3e, model structure in Supplementary Fig. 19). Notably, the intermolecular interactions of CX-Cellulose and CX-Lignin increased with the introduction of carboxylic acid moieties (Fig. 3f and Supplementary Fig. 20). To further validate these interactions in a more realistic model system, we expanded the DFT analysis to a periodic crystal model containing 20 molecules under periodic boundary conditions, simulating an extended bulk structure. The calculated total intermolecular interaction energy within the unit cell aligns with our initial results from smaller models, consistently confirming that the enhancement of intermolecular interactions in CX-Wood is a robust feature, which underpins its improved photophysical properties (Supplementary Fig. 21).

Additionally, transient absorption spectra of CX-Wood with DS = 0.3288 and DS = 0.5760 were determined. The obvious positive absorption bands located at 400-700 nm are ascribed to long-lived triplet states from CX-Wood (Supplementary Fig. 22). The gradual decrease in the positive absorption bands represents the continuous relaxation of CX-Wood from the triplet state to the ground state[67]. Furthermore, the kinetic decay curves at the peak for both types of CX-Wood with different degrees of substitution were analyzed under 340 nm excitation (Supplementary Fig. 23). Compared with the CX-Wood of lower substitution degree, the CX-Wood of higher substitution degree has a higher population rate of triplet excitons[68]. Meanwhile, we calculated the smallest non-radiative rate constants value ($k_{nr}^{Phos}$) and the fast intersystem crossing rate constant values ($k_{ISC}$) for natural wood and CX-Wood with different substitution degrees. After modification, CX-Wood possesses a longer lifetime and higher quantum yield compared to natural wood. At DS = 0.5760, the $k_{nr}^{Phos}$ for CX-Wood was 2.66 s$^{-1}$, significantly lower than the value of 86.29 s$^{-1}$ for natural wood. This indicates that the introduction of carboxymethyl groups substantially reduces the nonradiative decay rate in natural wood (Supplementary Table 3). Furthermore, the $k_{ISC}$ for CX-Wood consistently increases with increasing modification. Additionally, the calculated singlet-triplet energy gap ($\Delta E_{ST}$) shows a clear decrease with increasing substitution, providing a favorable thermodynamic driving force for the enhanced intersystem crossing[69] (Supplementary Table 4).

Interestingly, DFT calculations further suggested that the SOC value of lignin in the wood increased after decorating the structure with carboxylic moieties. This is because the n-π* electronic transition of the unsaturated C=O groups in the carboxyl moieties facilitates enhanced SOC and produces more triplet excitons through the subsequent ISC process[70], thereby boosting the room-temperature phosphorescence emission of CX-Wood (Supplementary Fig. 24).

These results, combined with the XRD and FT-IR pattern, suggest that chemical decoration enhanced the hydrogen bonding interactions and SOC in CX-Wood. This provides a more rigid environment and reduces $\Delta E_{ST}$, thereby promoting the population of triplet excitons, facilitating the intersystem crossing process, and ultimately enhancing the RTP emission.

## 3D printing via DIW

CX-Wood was dispersed in water to produce DIW inks. The rheologic properties of CX-Wood were then evaluated since it is crucial for the practical application. The rheological curves of CX-Wood ink were measured at different mesh sizes. When the wood particle size is coarser than 20 mesh, no intersection occurs between G′ and G″, indicating that CX-Wood fails to transition from a solid to a liquid state. This occurs because coarser particles result in insufficient reaction due to their larger size. As the mesh number increased, both the viscosity and substitution degree of CX-Wood rose with decreasing wood powder particle size (Fig. 4a and Supplementary Fig. 3). Meanwhile, the crossover modulus (G′ = G″) of CX-Wood decreased with increasing mesh number, indicating that CX-Wood ink is more likely to transform from a solid state to a liquid state at higher mesh counts. Based on a comprehensive evaluation of wood particle size effects on the modification efficiency, RTP performance, and rheological behavior of CX-Wood, we selected 60-mesh basswood powder for the research (Fig. 4b, Supplementary Figs. 3 and 25). Notably, the dispersion of CX-wood in aqueous solutions is not a simple particle system. Natural wood is composed of three primary components: cellulose, hemicellulose, and lignin. During the carboxylic modification process, the acid or base treatment partially disrupts the original supramolecular structures formed by these components, yielding either grafted polymer chains (resembling a "brush-like" structure) or free polymer chains[71]. These resulting structures engage in electrostatic and intermolecular interactions, which contribute to the high viscosity of the system. When shear stress is applied, these interactions are weakened, leading to a progressive decline in viscosity as shear rate increases. Thus, the shear-thinning behavior of CX-Wood is characteristic of pseudoplastic fluids.

For the 60-mesh CX-Wood inks, in the initial segments of the rheological curves, experimental results indicated that the apparent viscosity decreased with an increase in the shear rate, indicating that the as-prepared CX-Wood inks were pseudo-plastic fluids. Given that the inks possessed shear-thinning behavior, they are suitable for extrusion printing (Fig. 4a). Subsequently, the viscoelastic properties of CX-Wood, a key parameter for ensuring continuous printing, was analyzed through amplitude sweep, with the G′ and G″ verse shear stress plotted as shown in Fig. 4b. In the initial segments of the rheological curves, the ink exhibited solid-like elastic behavior. As the shear stress increases, G′ was lower than G″, indicating the transition of the ink into a liquid state. Notably, CX-Wood exhibited a relatively high G′ of 5.7 kPa at the yield stress and a low shear stress of 750 Pa at the flow point. Generally, CX-Wood inks can change from a non-flowing solid state to a flowable liquid state under low shear stress, so that the ink can be smoothly and continuously extruded from the nozzle, and then return to a solid-like state with strong elasticity after extrusion, keeping shape fidelity. These results clearly indicated that CX-Wood dispersion was suitable as a DIW ink[72].

Based on these results, CX-Wood was printed using a 3D printer. Notably, CX-Wood ink is formulated only from chemically modified wood powder (CX-Wood) and deionized water. The CX-Wood exhibited good adaptivity and could be printed using a printer nozzle with a wide range of sizes (Fig. 4c and Supplementary Movie 1). Upon hydration, hydrogen bonding interactions were formed between the -OH and -COOH in CX-Wood, determined by FT-IR spectra (Supplementary Fig. 2). SEM analysis revealed that as-formed hydrogen bonds promoted the association between these modified CX-wood powders, establishing a stabilized network and merged interface without any post cross-linking, and a porous structure was observed within the dried samples (Supplementary Fig. 26). Notably, the samples printed with CX-Wood have a certain structural shrinkage after drying, retaining ~90% of their original morphology (Supplementary Fig. 27). This was due to the collapse of the internal structure induced by the

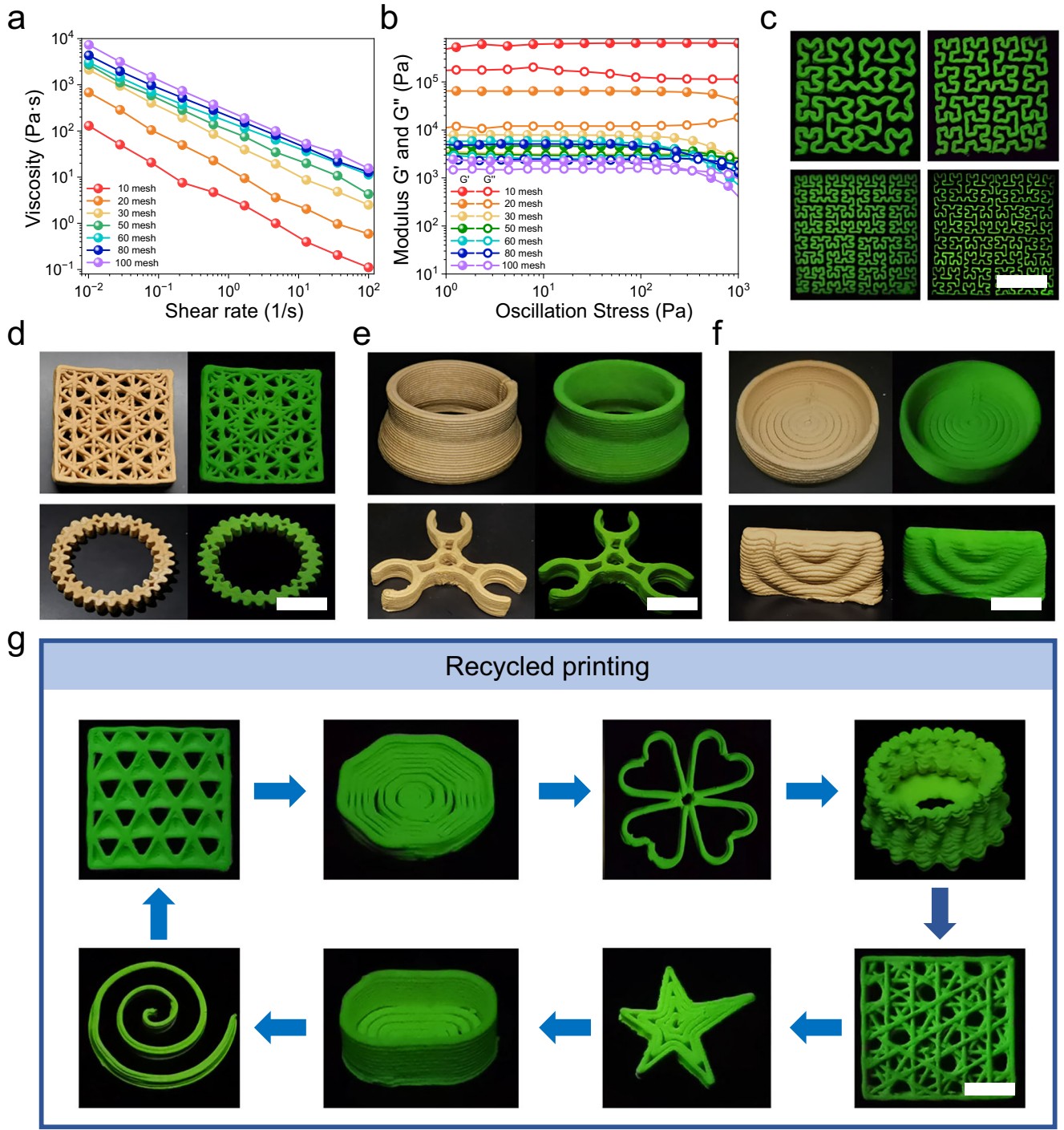

**Fig. 4 | DIW print performance of CX-Wood. a** The dependence of viscosity of CX-Wood on shear rate for different-sized particles. **b** Storage modulus (G′) and loss modulus (G″) plotted versus shear stress of CX-Wood for different-sized particles. **c** CX-Wood shapes printed at different printing accuracy, scale bar = 5 cm. **d**–**f** CX-Wood shapes printed using different wood species, scale bar = 2 cm. **g** Recyclable 3D printing of CX-Wood, scale bar = 2 cm.

capillary force resulting from the evaporation of water during the drying process[73]. Meanwhile, the material also exhibits excellent mechanical properties and flame retardancy, making it suitable for a wide range of applications (Supplementary Figs. 28–36).

 With these results in hand, the optical performance of the printed structures was determined. The as-obtained 3D shapes exhibited green afterglow RTP emission after switching off the UV excitation sources (Fig. 4d–f). Moreover, CX-Wood, made from eight types of wood, including balsawood, teakwood, poplar, pine, maple, beech, walnut,

and schima, could be printed using a DIW printer, confirming the generality of the method (Supplementary Fig. 37).

 Since the printed structures were not cross-linked, structural dissolution upon immersion in water was possible. Taking advantage of this property, the as-printed materials could be recycled for reuse. Specifically, the as-printed materials were converted into printable inks via water processing. The as-obtained inks could then be further processed into different 3D structures according to requirements.

Notably, such processing could be recycled several times, and the RTP emission properties of these materials were retained (Fig. 4g).

## Discussion

Natural wood was efficiently converted into CX-Wood via chemical modification with carboxylic acid moieties. The as-obtained CX-Wood exhibited RTP emission with a lifetime of 358.72 ms. Interestingly, the RTP emission from CX-Wood was general, and ~ 10 types of wood exhibited effective RTP emission after chemical modification with carboxylic acid moieties. Moreover, the favorable rheological properties of CX-Wood in aqueous solution enabled its use as a printable ink for DIW printing. As a result, a series of 3D RTP structures was flexibly designed and obtained using CX-Wood. Moreover, these obtained RTP structures could be recycled for 3D printing use several times via water treatment. Most available printable wooden materials can only be obtained by physically mixing wood with polymers, requiring expensive and complicated preparation procedures. As such, combined with the widespread application of printable wooden materials, CX-Wood represents a very promising source of structural materials for architecture, furniture, and other applications, which can be attributed to the sustainable, convenient, and cheap preparation of CX-Wood.

## Methods

### Preparation of CX-Wood

Wood (5 g) was added to an ethanol-water solution (100 mL, 85%) with NaOH (4 g). The mixture was then stirred at 35 °C for 60 min. After that, the reaction temperature was increased to 70 °C, and 25 mL of 85% ethanol-water solution containing 7 g of chloroacetic acid was added. The mixture was stirred for 30 minutes. After that, the reaction temperature was increased to 80 °C, 25 mL 85% ethanol-water solution containing 2 g NaOH was added. The reaction was carried out at 80 °C for 90 min. The as-obtained reaction mixtures were filtered through a glass filter and washed twice with 85% ethanol-water solution and then twice with anhydrous ethanol. The solid product was dried at 80 °C for 4 h to give CX-Wood.

### Preparation of CX-Cellulose

The raw material is microcrystalline cellulose. The preparation method and post-treatment are the same as those of CX-Wood.

### Preparation of CX-Lignin

The raw material is alkali lignin, and the preparation method is the same as that for CX-Wood. The post-treatment methods are as follows: To the post-reaction solution, hydrochloric acid was added dropwise to adjust the pH to 2 - 3, precipitating the lignin by capitalizing on its alkali-soluble/acid-insoluble properties. The precipitate was filtered and collected, washed twice with an 85% ethanol-water solution, and then twice with anhydrous ethanol. The solid product was dried at 80 °C for 4 h to give CX-Lignin.

### Preparation of CX-Wood inks

CX-Wood (100 g, 60 mesh) was added to deionized water (75 mL). Then, the mixture was mechanically stirred at room temperature for 30 minutes to give CX-Wood inks.

## Data availability

All relevant data are included in this article and its Supplementary Information files. Source data are provided with this paper. All data underlying this study are available from the corresponding author Yingxiang Zhai upon request. Source data are provided with this paper.

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

## Acknowledgements

This work was supported by the National Natural Science Foundation of China (32471803, Z.C.), Fundamental Research Funds for the Central Universities (2572022CG02, Z.C.), and the Open Research Fund of the School of Chemistry and Chemical Engineering, Henan Normal University (2020ZD01, T. D. J.). Thanks to Jinan Chenkun Intelligent Technology Co., Ltd. for providing technical support.

## Author contributions

Conceptualization: Z.C., S. Li., Y.Z., and T.D.J.; Methodology: Z.C., Y.Z.; Investigation: K.W., Y.Z.; Visualization: K.W., Y.B., Z.P., J.Z., M.W.; Supervision: Y.Z., L.W., X.L., C.Y., S. Liu., S. Li., J.L., S.C., Y.W. and T.D.J.; Writing-original draft: All authors; Writing-review and editing: All authors.

## Competing interests

The authors declare no competing interests.
