## [Transparent Peer Review file · Nature Communications]

3D-printable phosphorescent woody materials

Corresponding Author: Professor Tony D. James

Version 0:

Reviewer comments:

Reviewer #1

(Remarks to the Author)

The manuscript presents a study on the development of phosphorescent wood-based materials through the integration of natural phosphors and discusses their applicability in 3D printing. While the author presented the manuscript nicely and the subject matter aligns with sustainable functional materials, the manuscript lacks novelty and significance of the presented research.

The manuscript did not clearly discuss the specific research gap. While the authors briefly mention the limitations of existing luminescent materials, there is insufficient discussion on how this particular work significantly advances the state of the art. The paper should clearly differentiate this study from the existing body of work, especially those already published by the same authors and others. There are a lot of works on wood and other biomass resources that have been focused on developing room temperature phosphorescent (RTP) (a few of them are Nat. Commun. 14, 2614 (2023), Nat. Commun. 16, 868 (2025)). The author can claim that the novelty might lie in the 3D printing part. However, 3d printing of wood has also been reported (Science Advances 2024, 10, eadk3250).

The manuscript should justify why this particular research is significantly important compared to the present literature in terms of broader impacts, scalability, and unique functional properties. Simply combining phosphorescence with wood or bio-inks is not novel and sufficient unless a clear application-driven or performance-based advantage is shown. For example, wood is a structural material. In furniture and other building applications, the mechanical properties are most important. That's why the author must incorporate the mechanical or structural integrity of the developed materials, and I do not think the developed material using the direct write process can compete with natural wood from a structural aspect. Without such data, the functionality and reliability of the material in practical applications remain uncertain.

Considering the novelty and significance of the research, I do not recommend publishing this manuscript in a prestigious journal like Nature Communications.

Reviewer #2

(Remarks to the Author)

This manuscript has reported chemical modification of natural wood to introduce carboxymethyl groups. The modified wood powder indicated green phosphorescence with a lifetime in the range of hundreds of ms. And it is used for direct ink writing to manuscript different morphologies in the scale of macroscopic level. The entire work is more focused on engineering instead of addressing scientific problems. And the entire manuscript only presented four figures, one of them is schematic illustration. Even so, the supporting information also has limited data. All the data presented in the manuscript is unable to fully understand the mechanism of the phosphorescence of the carboxymethylated wood, and the advantage of the 3D printing of the wood. At this stage, this work is not suitable to be published in Nature Communications.

1. What is the advantage and novelty of the chemical reaction utilized here? As known, the chemical modification of cellulose to be carboxymethylated is a well-developed reaction. And, carboxymethyl cellulose is even commercially available.

2. In comparison to previous literature, quite many other chemical modifications have been reported to modify the cellulose for obtaining phosphorescence, such as Chemical Engineering Journal 471 (2023) 144665, Nat Commun 13, 1117 (2022), Adv. Funct. Mater. 2024, 34, 2403977, Adv. Mater. 2023, 35, 2304032. The RTP performance in the reported work can reach a lifetime above 1 seconds, and tunable RTP emission colors, as well as stimulus responsive RTP. Thus, in the aspect of

RTP performance, the advantage of current work is not obvious.

3. The mechanism of RTP from current carboxymethylated wood could have many explanations. Current data does not address this. In line 148, the authors do mention that “enhancing hydrogen bonding interactions within cellulose”, and in line 160, “enhanced the hydrogen bonding interactions in CX-Wood, providing a more rigid environment and, thus, promoting the RTP emission.” In the current system, there is no phosphor added, so the hydrogen bonding formed between the cellulose. How does the formation of H-bonding enhance RTP? Whether the population of the triplet state excitons is increased, or the intersystem crossing is enhanced. This should be clearly explained based on experimental data. The authors are suggested to utilize transient absorption spectroscopy with a temporal resolution in the range of femtosecond to explore this question.

4. For the DFT simulation, it is usually used for small systems. In current work, the macromolecules of cellulose/hemicellulose, and the lignin are quite complicated. Whether the simulation can represent the experimental situation is questionable. And there is no detailed setting of the simulation in the paper.

5. How is the quantum yield of the RTP as a function of the carboxymethylated degree?

6. Figure 4e indicated an excitation wavelength dependent shift of the emission. Any idea about which groups of the emission peaks are assigned, correspondingly?

7. As concerning the 3D printing part, what are the challenges the authors have conquered, and based on what kind of strategy is not clear presented? Or just by mixing with water, and adjusting the concentration and viscosity? If it is by adjusting these parameters, the novelty of 3D printing of wood powders is not good enough.

8. There are papers have published about 3D printed carboxymethyl cellulose, such as Carbohydrate Polymers, 278, 2022, 118924 (doi.org/10.1016/j.carbpol.2021.118924.), and summarized in the reviewer paper (Mater. Chem. Front., 2022, 6, 254–279). The authors need to clearly address the scientific advantage of the 3D print of the carboxymethylated wood.

9. According to the experimental section, the 3D printed wood powder is not post-cross-linked. Did the ink only contain wood and water? It is not clearly stated.

10. As the water evaporated after printing, will the structure shrink? If so, shrinking to what degree? And how well can the morphology be reserved? Are any porous structures formed?

11. How are the mechanical properties, such as elastic modulus, tensile stress, and compress stress of the 3D printed structures?

12. Is the natural wood grounded into powder? What is the size of the powder? Will the size affect the chemical modification degree?

13. If the modified RTP wood is in powder manner, how does the size affect the rheological behaviors of the wood ink?

14. Figure 2g, the mark of 80 °C is confused, tested at the temperature, or the wood is heated at 80°C to remove the water? But this temperature is not supposed to remove all the water.

15. The discussion in 2.1 is suggested to be well-developed, currently, most of the writing is simply describing the phenomenon, instead of developed explanations. And many references are missing about all the xps, ftir peak assignments.

Reviewer #3

(Remarks to the Author)

The authors reported a concise strategy to construct room-temperature phosphorescence (RTP) woody materials by grafting carboxyl-functional groups onto native lignocellulosic matrices, endowing the RTP wood with the property of direct ink writing. The modified wood exhibited intense and long-lived RTP with a lifetime of 358.7 ms. The strategy is effective for the majority of woods. Impressively, a series of 3D RTP structures were flexibly designed and obtained using CX-Wood, and these obtained RTP structures could be recycled for 3D printing via water treatment. These RTP woods demonstrate a very promising source of structural materials for architecture, furniture and other applications. The results are of considerable interest, and potentially merit publication as a Nature Communication. However, there are a number of important points which must be addressed:

1.As shown in Figure 1a, the modified wood is coated with sodium acetate ions, I wonder whether the ionic bonding also plays an important role in generating RTP?

2.In Figure 2e, CX-Wood exhibited excitation-dependent RTP emission. Please explain the reason for this phenomenon

3.The quantum yields of CX-Wood should be added.

4.The mechanical strength of wood before and after the modification should be measured, such as compressive stress–strain curves.

5.For the preparation of different degree of substitution (DS) determination of CX-Wood, more experiment details should be given. Whether different degrees of substitution (DS) determination of CX-Wood depend on the reaction time or temperature?

Version 1:

Reviewer comments:

Reviewer #1

(Remarks to the Author)

I am not still convinced about the response on this manuscript. The response did not clearly discuss my concern regarding the novelty aspect and this is still not be considered as new concept. Rather, they have provided a lot of unwanted testing. Also, very recently the author published a similar article in Nature Communications (Nat Commun 16, 7978 (2025)). Again, there are a lot of works on wood and other biomass resources that have been focused on developing room temperature phosphorescent (RTP) (Nat. Commun. 14, 2614 (2023), Nat. Commun. 16, 868 (2025)). The author claimed that the novelty

lied in the 3D printing part. However, 3d printing of wood has been reported (Science Advances 2024, 10, eadk3250). As I have seen in the paper from Thakur et al., they also did not use any additives. They printed wood from all-wood components. So, the author's claim of additive-free for the first time is not also correct. Considering the novelty and significance of the research, I do not recommend publishing this manuscript in Nature Communications.

Reviewer #2

(Remarks to the Author)

Thanks to the authors for the efforts in addressing the concerns raised. However, several key issues remain insufficiently resolved.

1. One of the main questions is the novelty of the work as mentioned in Question 1. Unfortunately, the response appears to miss the core issue. Rather than addressing the novelty directly, the authors primarily compare CX-wood to CMC-based materials. The mention of CMC in the original question was intended to highlight that the chemical modification of wood to introduce carboxymethyl groups is a well-established and even commercialized technique. Therefore, the carboxymethylation of wood cannot be considered a novelty in this context.

2. The second claimed novelty is the 3D printability of CX-wood without additional additives—simply by mixing with DI water. However, to facilitate this, what kind of scientific problem the authors have solved by developing what kind of new method? It seems the printability is facilitated by grafting the carboxymethyl groups. But this is nothing new.

3. Moreover, for the view of printable phosphorescence wood or cellulose materials, current system also does not present an obviously advancement. The manuscript introduces limited new insights or strategies to achieve RTP, as quite many previous work reported in this field, to name a few Nat. Commun. 14, 2614 (2023), Nat. Commun. 16, 868 (2025), Adv. Mater. 2023, 35, 2304032, Adv. Funct. Mater. 2024, 34, 2403977.

Give these, the revised manuscript does not demonstrate a clear improvement in terms of novel findings or methodological innovations that effectively address a specific scientific gap in the field.

Based on this, the selling points of current work is not strong. And the novelty declared here is not convincing. Therefore, it is not suggested to be published in Nature Communications.

And few more concerns as following:

4. The discussion and presented data in the manuscript is missing a focal point. Although the authors declare the point of current work is the 3D printing of CX-wood. However, more than two-thirds of the results and discussion of the manuscript is devoted to the phosphorescence, and two out of four main figures exclusively present phosphorescence data. The remaining one-third discussion covers multiple properties of the printed structure, such as mechanical properties, flame-retardant properties, decomposition etc. The layout and data presentation result in a scattered manuscript lacking a clearly defined focal point.

5. The 3D print ink is composed of 60 mesh-size CX-wood micro-particle dispersed in DI water. The ink is not a polymer solution, but a particle-based suspension. This kind of fluid is usually a dilatant liquid, exhibiting shear-thickening (dilatant) behavior, in which shear stress increases with higher shear rates. However, the rheological property characterized here show the property of pseudoplastic fluid. The authors are suggested to explain and dig deep about this.

6. The authors compared the mechanical properties of the printed structures based on CX-wood and CMC. The mechanical property is depended on the porous structure. It is not reasonable to compare this without talking about the internal morphology of the printed structures.

7. Similarly, for the flame-retardant property, which is typical feature for materials with porous structure as reported in many aerogel materials, for instance, Chemical Engineering Journal 499 (2024) 155939. The flame-retardant effect observed here may stem from porosity or other physical factors rather than only intrinsic material properties. The authors are encouraged to explore and clarify the underlying mechanism.

Reviewer #3

(Remarks to the Author)

I agree to the revised version.

Version 2:

Reviewer comments:

Reviewer #2

(Remarks to the Author)

In the revised version, the authors have not made substantial changes to the manuscript either about the novelty or the results. Instead, they mainly added further explanations/augments about the claimed novelty. In the revised version, there is no strong evident or data to significantly enhance the scientific importance of the manuscript. The newly added data are a life cycle assessment (LCA) methodology and a DFT simulated spin orbit coupling (SOC) values, both presented in the supporting information.

The former one could be served as an add-on, industry-oriented consideration, but it does not strengthen the scientific novelty of the study. Regarding the DFT results, the phosphorescent emission inherently involves intersystem crossing from the singlet to the triplet state, a process that is universally governed by SOC, which is not specific to the present system. Moreover, the enhance of lifetime may originate from multiple factors. It is risk to ascribe the reason to the enhanced SOC value, and the value is obtained by simulation.

Moreover, as mentioned in the previously round of peer-review, both the 3D printing of wood and the realization of room

temperature phosphorescence (RTP) in wood have been reported individually in previously publications. In revised work, the authors declare the novelty based on combining these two aspects to fabricate 3D-printed wood-based materials exhibiting RTP. To enable this functionality, the authors employ carboxymethylation of wood. However, this is a well-established and widely used chemical modification method for wood-based materials. There is not concept or strategy developed here. On the other hand, as concerning the RTP performance, the lifetime is on the order of hundreds of ms. In the field of organic RTP materials, this performance is relatively ordinary. The reported lifetime has been in the range of seconds even in the publications reported couple of years ago. Thus, the novelty, the development of a new scientific concept or methodological strategy, and the RTP performance of the revised manuscript are not sufficiently convincing. It is not suggested to be published in Nature Communications.

Reviewer #3

[Editorial Note: Reviewer #3 was asked to look over the responses to Reviewer #2]

(Remarks to the Author)

Since single-crystal structures cannot be obtained for this material system, specific intermolecular interactions and molecular packing arrangements remain undetermined. The theoretical calculations presented in the article are all based on computational simulations, and their confirmatory value is open to debatable. Only by performing simulations on a large number of molecules can the results approach closer to the real situation. Therefore, the accuracy of the DFT (Density Functional Theory) calculations is subject to further discussion.

Phosphorescent wood reported typically involved creating models through 3D printing and then soaking them in phosphorescent dyes. This work proposes directly printing modified wood into 3D models, ensuring that both the interior and exterior of the models consist of phosphorescent material. I believe this work has significant potential to advance the industrialization of phosphorescent materials. I agree to the revised version.

Version 3:

Reviewer comments:

Reviewer #3

(Remarks to the Author)

The revisions have resolved my issues, and I agree to publish the work.

Response to the reviewers

Reviewers' comments:

Reviewer #1 (Remarks to the Author):

Reviewer 1: The manuscript presents a study on the development of phosphorescent wood-based materials through the integration of natural phosphors and discusses their applicability in 3D printing. While the author presented the manuscript nicely and the subject matter aligns with sustainable functional materials, the manuscript lacks novelty and significance of the presented research.

The manuscript did not clearly discuss the specific research gap. While the authors briefly mention the limitations of existing luminescent materials, there is insufficient discussion on how this particular work significantly advances the state of the art. The paper should clearly differentiate this study from the existing body of work, especially those already published by the same authors and others. There are a lot of works on wood and other biomass resources that have been focused on developing room temperature phosphorescent (RTP) (a few of them are Nat. Commun. 14, 2614 (2023), Nat. Commun. 16, 868 (2025)). The author can claim that the novelty might lie in the 3D printing part. However, 3d printing of wood has also been reported (Science Advances 2024, 10, eadk3250).

The manuscript should justify why this particular research is significantly important compared to the present literature in terms of broader impacts, scalability, and unique functional properties. Simply combining phosphorescence with wood or bio-inks is not novel and sufficient unless a clear application-driven or performance-based advantage is shown. For example, wood is a structural material. In furniture and other building applications, the mechanical properties are most important. That's why the author must incorporate the mechanical or structural integrity of the developed materials, and I do not think the developed material using the direct write process can compete with natural wood from a structural aspect. Without such data, the functionality and reliability of the material in practical applications remain uncertain.

Considering the novelty and significance of the research, I do not recommend

publishing this manuscript in a prestigious journal like Nature Communications.

A: Thanks for the comments.

Indeed, biomass-derived RTP materials were reported. Producing RTP materials from biomass sources, particularly from natural wood, represents a sustainable trend considering the abundance, low cost, renewable and inherent RTP emission of wood. As a result, lots of research has focused on this area. However, shaping these woody RTP materials primarily relies on subtractive manufacturing techniques. Such processes often generate substantial waste, leading to material loss and increased production costs. Thus, we mixed modified wood and thermoplastic PP to successfully create a 3D printable woody RTP materials (*Nat. Commun.*, 2023, 14, 2614). Nevertheless, the whole preparation of 3D printable woody RTP materials involves multi-step processes and petrol-derived resources.

Additionally, among the various AM techniques for wood, direct ink writing (DIW) has emerged as the most versatile 3D printing technique since it acts as a unique pathway to introduce design freedom, multifunctionality, and stability simultaneously into the printed structures. Generally, DIW requires the appropriate rheological behavior of the printed inks. Thus, wood is usually engineered to exhibit the appropriate rheological behavior for printing by introducing thermoplastic plastics (such as PE, PP, PVC, PLA and so on) (*Compos. B Eng.*, 2020, 200, 108347; *Compos. B Eng.*, 2021, 219, 108873; *Adv. Compos. Hybrid Mater.*, 2023, 6, 81; *Adv. Drug Deliv. Rev.*, 107, 2016, 17-46). Nevertheless, these strategies for converting wood into DIW inks either require complicated procedures or petrol-derived resources.

Additionally, the DIW inks can also be prepared from wood by extracting components (cellulose, hemicellulose and lignin) and subsequent modification for achieving the appropriate rheological properties. For example, Thakur et al. reported on the introduction of nanocellulose into woody components to tune the rheological behavior for creating DIW inks (*Sci. Adv.*, 2024, 10, eadk3250). Such research represents substantial progress. However, using wood directly in the absence of additives for DIW printing still remains a challenge.

All these shortcomings motivated us to create a more convenient and sustainable 3D

printable RTP formula using wood. Here, we tuned the rheological behavior of wood by simply modifying the wood with carboxylic moieties. Our as-obtained CX-Wood exhibited suitable rheological behavior for DIW printing.

We measured the mechanical properties of the structure. Generally, the as-obtained 3D structures from CX-Wood exhibited good mechanical performance and with some improvement compared with natural wood in transverse tensile strength and longitudinal compressive strength (**Supplementary Fig. 25**). Specifically, we compared the mechanical strength between CX-Wood and natural woods (basswood and balsawood). Due to the anisotropic nature of natural woods (*Nat. Rev. Mater.*, 2020, 5, 642-666), they exhibit different tensile strengths in the "T-the radial direction of the wood", "R-the tangential direction of the wood", and "L-the longitudinal direction of the wood". Natural basswood exhibits strong tensile performance only in the "L" direction, with tensile strength and Young's modulus of 70.6 MPa and 547.3 MPa, respectively. However, its tensile properties in both the T- and R-directions are inferior to those of CX-Wood. Natural balsa wood, in all three directions (T, R, and L), demonstrates lower tensile performance than CX-Wood.

Regarding compressive strength, both basswood and balsawood exhibit strong compressive performance in the "TR" direction. Under a pressure of 2000 N, basswood exhibits compressive strength and compressive modulus of 8.4 MPa and 169 MPa respectively in the "TR" direction, while balsawood shows values of 6.1 MPa and 126.2 MPa in the same direction. However, the compressive properties of both woods in the "LR" direction are inferior to those of CX-Wood.

In addition, a dispersion of CX-Wood in an aqueous solution of polyurethane could be used as inks for direct printing. The as-printed structure was then dried by heating. Meanwhile, the post-crosslinking reaction between CX-Wood and polyurethane occurred during the process. The as-obtained structure exhibited good flexibility and retained its integrity after five compression cycles, whereas natural wood underwent structural collapse after just a single compression (**Supplementary Fig. 26**).

Moreover, as-printed materials from CX-Wood exhibited improved flame-retardant properties when compared with natural wood, which was also important for its

application for furniture and other building purposes. Specifically, the thermogravimetric analysis (TG) and differential thermogravimetric analysis (DTG) curves in the continuous and trigger modes indicated that CX-Wood generated more residues than natural wood (**Supplementary Fig. 29**). This is because under high-temperature conditions, the CX-wood undergoes carbonization, a process catalyzed by sodium ions. This stable carbonate barrier layer acts as a barrier between the air and the interior of the CX-Wood, thereby preventing the spread of flames (*Carbohydr. Polym.*, 2017, 157, 1594-1603). The maximum LOI value of CX-Wood was 39.5%, which is higher than that (19.8%) for natural wood (**Supplementary Fig. 30**). Vertical burning tests indicated that natural wood was flammable and there was no rating according to the UL-94 standard. In comparison, CX-Wood exhibited superior self-extinguishing performance after two ignitions (fire treatment for 10 s each time), reaching the highest level (V-0 grade, UL-94 standard) (*Prog. Polym. Sci.*, 2017, 69, 22-46) (**Supplementary Fig. 31**). The CONE results indicated that the values of heat-release rate, total heat release, and the total smoke production for CX-Wood were significantly lower than natural wood. The residual char of CX-Wood reached 39.7% at the end of the CONE tests, and there was almost no residue for natural wood. Further analysis of the carbon residue results from CC tests of natural wood and CX-Wood revealed that the alkaline environment induced by Na⁺ in CX-Wood facilitated dehydration reactions, enabling rapid formation of continuously growing carbon layers and thereby enhancing its flame retardant nature (**Supplementary Fig. 32**).

Generally, CX-Wood exhibited advantages in the functionality and reliability. In the previous version of our manuscript, we did not clarify these important advantages clearly. Therefore, we have now, added these adventitious results for CX-Wood to the revised manuscript.

Supplementary Fig. 25. (a) Schematic diagram of Tensile and compression testing directions for natural wood (basswood and balsawood). (b) Tensile (tensile strength and Young's modulus) properties of CX-Wood and basswood and balsawood. (c) Compressive (compress strength and compress modulus) properties of CX-Wood and basswood and balsawood (the pressure is 2000 N).

Supplementary Fig. 26. (a) Compression images of natural wood and CX-Wood/Polyurethane. (b) Five-cycle compression curves of CX-Wood/Polyurethane.

Supplementary Fig. 29. (a) TG and (b) DTG curves for Natural wood and CX-Wood in nitrogen atmosphere.

Supplementary Fig. 30. The limiting oxygen index of Natural wood and CX-Wood.

Supplementary Fig. 31. Vertical burning of Natural wood and CX-Wood (UL-94 test).

Supplementary Fig. 32. Cone calorimeter of Natural wood and CX-Wood. (a) HRR curves. (b) THR curves. (c) TSP curves. (d) Mass retention curves. (e) SEM images of the residual char of Natural wood. (f) SEM images of the residual char of CX-Wood.

Reviewer #2 (Remarks to the Author):

Reviewer 2: This manuscript has reported chemical modification of natural wood to introduce carboxymethyl groups. The modified wood powder indicated green phosphorescence with a lifetime in the range of hundreds of ms. And it is used for direct ink writing to manuscript different morphologies in the scale of macroscopic level. The entire work is more focused on engineering instead of addressing scientific problems. And the entire manuscript only presented four figures, one of them is schematic illustration. Even so, the supporting information also has limited data. All the data presented in the manuscript is unable to fully understand the mechanism of the phosphorescence of the carboxymethylated wood, and the advantage of the 3D printing of the wood. At this stage, this work is not suitable to be published in Nature Communications.

Thanks for the constructive comments.

1. What is the advantage and novelty of the chemical reaction utilized here? As known, the chemical modification of cellulose to be carboxymethylated is a well-developed reaction. And, carboxymethyl cellulose is even commercially available.

A: Thanks for the comments. In this article, we converted wood into RTP materials suitable for DIW printing through a carboxymethylation method.

Compared with commercial carboxymethyl cellulose, CX-Wood shows the following advantages.

(1) In terms of manufacturing process, unlike carboxymethyl cellulose, our method eliminates the need for cellulose extraction from bulk wood, making the process more straightforward.

(2) Additionally, we compared the differences in printing performance between CX-Wood and commercial carboxymethyl cellulose. Due to the presence of lignin in CX-Wood, which provides adhesive and structural support, the printed samples exhibited self-standing shape, whereas the structures printed using commercial carboxymethyl cellulose are prone to collapse (*Adv. Mater.*, 2021, 33, 2001588) (**Supplementary Fig. 27**).

(3) The tensile strength of the printed structures from commercial CMC are also inferior to those of CX-Wood (**Supplementary Fig. 28**).

(4) Moreover, CX-Wood demonstrates excellent flame-retardant properties. The thermogravimetric analysis (TG) and differential thermogravimetric analysis (DTG) curves in the continuous and trigger modes indicated that CX-Wood generated more residues than natural wood (**Supplementary Fig. 29**). This is because under high-temperature conditions, the CX-Wood undergoes carbonization, a process catalyzed by sodium ions. The stable carbonate barrier layer acts as a barrier between the air and the interior of the CX-Wood, thereby preventing the spread of flames (*Carbohydr. Polym.*, 2017, 157, 1594-1603). The maximum LOI value of CX-Wood was 39.5%, which is higher than that (19.8%) for natural wood (**Supplementary Fig. 30**). Vertical burning tests shows that the natural wood was flammable and there was no rating according to the UL-94 standard. As comparison, CX-Wood exhibited superior self-extinguishing performance after two ignitions (fire treatment for 10 s every time), reaching the highest level (V-0 grade, UL-94 standard) (*Prog. Polym. Sci.*, 2017, 69, 22-46) (**Supplementary Fig. 31**). The CONE results indicated that the values of heat-release rate, total heat release, and the total smoke production for CX-Wood were significantly lower than natural wood. The residual char of CX-Wood reached 39.7% at the end of the CONE tests, and there was almost no residue for natural wood. Further analysis of the carbon residue results from CC tests of natural wood and CX-Wood revealed that the alkaline environment induced by Na⁺ in CX-Wood facilitated dehydration reactions, enabling rapid formation of continuously growing carbon layers and thereby enhancing its flame retardancy (**Supplementary Fig. 32**).

We added the results to the revised manuscript as **Supplementary Fig. 27**, **Supplementary Fig. 28**, and **Supplementary Fig. 29-32**.

Commercial CMC

CX-Wood

Supplementary Fig. 27. Samples printed using commercial CMC and CX-Wood. As shown in the figure, the samples printed with CX-Wood retain their structural integrity, while those printed with commercial CMC exhibit varying degrees of collapse (highlighted in red).

Supplementary Fig. 28. Tensile strength of the structure printed from commercial CMC and CX-Wood.

Supplementary Fig. 29. (a) TG and (b) DTG curves for Natural wood and CX-Wood in a nitrogen atmosphere.

Supplementary Fig. 30. The limiting oxygen index of Natural wood and CX-Wood.

Supplementary Fig. 31. Vertical burning of Natural wood and CX-Wood (UL-94 test).

Supplementary Fig. 32. Cone calorimeter of Natural wood and CX-Wood. (a) HRR curves. (b) THR curves. (c) TSP curves. (d) Mass retention curves. (e) SEM images of the residual char of Natural wood. (f) SEM images of the residual char of CX-Wood.

2. In comparison to previous literature, quite many other chemical modifications have been reported to modify the cellulose for obtaining phosphorescence, such as Chemical Engineering Journal 471 (2023) 144665, Nat Commun 13, 1117 (2022), Adv. Funct. Mater. 2024, 34, 2403977, Adv. Mater. 2023, 35, 2304032. The RTP performance in the reported work can reach a lifetime above 1 seconds, and tunable RTP emission colors, as well as stimulus responsive RTP. Thus, in the aspect of RTP performance, the

advantage of current work is not obvious.

A: Thanks for the comments.

Indeed, the advantage of current RTP performance of CX-Wood was not obvious. However, the easy preparation and DIW printable properties enables CX-Wood with design freedom.

We found that introducing borax into the printed materials, further increases the rigidity of the system, and obviously increases the RTP lifetime. The lifetime of the modified CX-Wood can reach up to 1.06 s, and at the same time, a green afterglow lasting for up to 8 s can be observed with the naked eye (**Supplementary Fig. 12**).

Moreover, RhB can also be introduced into CX-Wood. The RTP emission of CX-Wood overlaps with the absorbance of RhB, so that red afterglow emission can be observed from CX-Wood via a triplet state to singlet state Förster resonance energy transfer (TS-FRET) strategy. The as-obtained CX-Wood/RhB exhibited a RTP emission at 605 nm, which was consistent with its fluorescence spectrum. The lifetime of red afterglow emission from RhB in the CX-Wood/RhB was 78.08 ms (**Supplementary Fig. 13**).

For stimuli-responsive RTP emission, CX-Wood exhibits sensitivity to temperature. Both the RTP intensity and lifetime decreased upon increasing the temperature. Specifically, the lifetime of CX-Wood decreased from 500.37 ms to 36.14 ms when the temperature increased from 77 K to 400 K (**Supplementary Fig. 10**).

We added the results to the revised manuscript as **Supplementary Fig. 12**, **Supplementary Fig. 13** and **Supplementary Fig. 10**.

Supplementary Fig. 12. Photographs of four-leaf clover printed by CX-Wood and CX-Wood + Borax after turning off a 365 nm UV lamp. (b) Afterglow emission spectra of CX-Wood and CX-Wood + Borax. (c) RTP decay profiles of CX-Wood and CX-Wood + Borax. (Preparation of CX-Wood + Borax: A 200 mg/mL borax solution was prepared. CX-Wood was then mixed with the borax solution to achieve 75% solid content. The mixture was dried at 80°C to obtain CX-Wood + Borax.)

Supplementary Fig. 13. (a) The RTP emission of CX-Wood/RhB and the absorbance of RhB. (b) Fluorescence (prompt) and phosphorescence (delay) spectra of B-film/RhB. (c) Lifetime decay of delayed fluorescence of B-film/RhB at 605 nm. Excitation wavelength = 340 nm.

Supplementary Fig. 10. Phosphorescence properties of CX-Wood under conditions of different temperatures. (a) Afterglow emission spectra of CX-Wood at different temperatures. (b) Lifetime decay profiles of CX-Wood at different temperatures. Excitation wavelength = 340 nm.

3. The mechanism of RTP from current carboxymethylated wood could have many explanations. Current data does not address this. In line 148, the authors do mention that “enhancing hydrogen bonding interactions within cellulose”, and in line 160, “enhanced the hydrogen bonding interactions in CX-Wood, providing a more rigid environment and, thus, promoting the RTP emission.” In the current system, there is no phosphor added, so the hydrogen bonding formed between the cellulose. How does the formation of H-bonding enhance RTP? Whether the population of the triplet state excitons is increased, or the intersystem crossing is enhanced. This should be clearly explained based on experimental data. The authors are suggested to utilize transient absorption spectroscopy with a temporal resolution in the range of femtosecond to explore this question.

A: Thanks for the comments. We tested the transient absorption spectra of CX-Wood with DS = 0.3288 and DS = 0.576. The obvious positive absorption bands located at 400-700 nm are ascribed to long-lived triplet states from CX-Wood. The gradual decrease in the positive absorption bands represents the continuous relaxation of CX-Wood from the triplet state to the ground state (*Angew. Chem. Int. Ed.*, 2024, 63, e202319089) (**Supplementary Fig. 20**). Furthermore, the kinetic decay curves at the

peak for both types of CX-Wood with different degrees of substitution were analyzed under 340 nm excitation. Compared with the CX-Wood of lower substitution degree, the CX-Wood of higher substitution degree has a higher population rate of triplet excitons (*J. Mater. Chem. C*, 2022, 10, 17620) (**Supplementary Fig. 21**).

Meanwhile, we calculated the smallest non-radiative rate constants value (k_{nr}^{Phos}) and the fast intersystem crossing rate constant values (K_{ISC}) for natural wood and CX-Wood with different substitution degrees. After modification, CX-Wood possesses a longer lifetime and higher quantum yield compared to natural wood. At DS=0.576, the k_{nr}^{Phos} for CX-Wood was 2.65 s^{-1} , significantly lower than the value of 86.29 s^{-1} for natural wood. This indicates that the introduction of carboxymethyl groups substantially reduces the nonradiative decay rate in natural wood (**Table S2**). Furthermore, the K_{ISC} for CX-Wood also consistently increases with increasing modification.

We added the results as **Supplementary Fig. 20**, **Supplementary Fig. 21**, and **Table S2**.

Supplementary Fig. 20. (a) Transient absorption two-dimensional temporal evolution

spectra at room temperature of CX-Wood with DS = 0.3288. (b) Fs-TA spectra of CX-Wood with DS = 0.3288 under 340 nm laser excitation. (c) Transient absorption two-dimensional temporal evolution spectra at room temperature of CX-Wood with DS = 0.576. (d) Fs-TA spectra of CX-Wood with DS = 0.576 under 340 nm laser excitation.

Supplementary Fig. 21. Transient absorption traces of CX-Wood with DS = 0.3288 at 650 nm and CX-Wood with DS = 0.576 at 640 nm.

Table S2 Photophysical properties of Natural wood and CX-Wood.

	DS	τ_F (ns)	τ_{RTP} (ms)	Φ_{RTP} (%)	$^a k_{ISC}$ (s^{-1})	$^b k_{nr}^{Phos}$ (s^{-1})
Natural wood	0	3.59	11.56	0.25	6.96×10^5	86.29
	0.0664	3.83	35.47	0.93	2.43×10^6	27.93
CX-Wood	0.1009	4.43	89.44	1.55	3.50×10^6	11.01
	0.3288	5.00	245.85	3.86	7.72×10^6	3.91
	0.576	5.43	358.72	4.60	8.47×10^6	2.66

a) $k_{ISC} = \Phi_{RTP}/\tau_F$; b) $k_{nr}^{Phos} = (1 - \Phi_{RTP})/\tau_{RTP}$. (*Sci. China Chem.*, 2023, 66, 1161-1168)

4. For the DFT simulation, it is usually used for small systems. In current work, the macromolecules of cellulose/ hemicellulose, and the lignin are quite complicated. Whether the simulation can represent the experimental situation is questionable. And there is no detailed setting of the simulation in the paper.

A: Thanks for the comments. Density Functional Theory as a contemporary computational method, is applicable for addressing medium-sized systems containing dozens to thousands of atoms (*Adv. Mater.*, 2023, 35, 2304032; *Adv. Opt. Mater.*, 2024, 12, 2401419.). In this research, we employed model structures of cellulose/lignin and their carboxymethyl-modified derivatives as computational models (**Supplementary Fig. 17** and **Supplementary Fig. 18**). The calculated results align with phosphorescence testing data, and the carboxymethyl modified cellulose/lignin has a stronger interaction force and better RTP performance at the same time, indicating that the simulation can reflect the experimental situation to a certain extent.

Additionally, for the section on DFT calculations we provide a detailed description as follows: “All quantum chemical calculations were performed using the Gaussian 16 software package (*Gaussian 16, Revision A.03, 2016*). The Becke, three-parameter, Lee–Yang–Parr (B3LYP) hybrid functional (*Phys. Rev. A*, 1988, 38, 3098) was employed throughout, in conjunction with the D3(BJ) dispersion correction (*J. Chem. Phys.*, 2010, 132, 154104) to account for long-range van der Waals interactions. For all geometry optimizations, the def2-SVP basis set (*J. Chem. Phys.*, 2001, 115, 9113) was chosen, which offers a good balance between computational cost and accuracy for systems of moderate size. The independent gradient model (IGM, **Supplementary Fig. 19**) wavefunction analysis was carried out using the Multiwfn program (*J. Chem. Phys.*, 2024, 161, 082503). Upon completion of the optimizations, the resulting molecular geometries and relevant properties were visualized and analyzed using Visual Molecular Dynamics (VMD) software (*J. Mol. Graph.*, 1996, 14, 33-38). The binding energy is calculated using the following formula:

$$E_{\text{bind}} = E_{\text{complex}} - (E_{\text{A}} + E_{\text{B}})$$

Where E_{complex} is the total energy of the complex, and E_{A} , E_{B} are the total energies of the individual components calculated separately.”

We added this information to the methods in the revised manuscript as **Supplementary Fig. 17, Supplementary Fig. 18 and Supplementary Fig. 19.**

Supplementary Fig. 17. The model structures of Cellulose and CX-Cellulose used for calculations.

Supplementary Fig. 18. The model structures of Lignin and CX-Lignin used for calculations.

Supplementary Fig. 19. The independent gradient model (IGM) analysis. In the IGM plot, regions with darker colors signify stronger interactions. Specifically, the blue and voluminous regions represent Coulombic interactions, while the green and flat regions denote dispersion interactions.

5. How is the quantum yield of the RTP as a function of the carboxymethylated degree?

A: Thanks for the comments. We have evaluated the quantum yields of CX-Wood with different degrees of substitution.

The phosphorescence quantum yield of the compound was obtained from the following equation:

$$\phi_{phos.} = \frac{B}{A} \times \phi_{PL}$$

Where A and B represent the integral areas of total photoluminescence and phosphorescence spectra, respectively. The phosphorescence was separated from PL by time-gating (*Nat. Mater.*, 2021, 20, 1539-1544).

The quantum yields of natural wood, CX-Wood with DS = 0.0664, CX-Wood with DS = 0.1009, CX-Wood with DS = 0.3288 and CX-Wood with DS = 0.5760 were measured to be 0.47%, 1.78%, 2.70%, 6.61%, and 8.32%, respectively. The phosphorescence quantum yield of natural wood, CX-Wood with DS = 0.0664, CX-Wood with DS = 0.1009, CX-Wood with DS = 0.3288 and CX-Wood with DS = 0.5760 were measured to be 0.25%, 0.93%, 1.55%, 3.86%, and 4.60%, respectively.

We added these results to the revised manuscript as **Supplementary Fig. 8**.

Supplementary Fig. 8. The quantum yields and phosphorescence quantum yields of natural wood and CX-Wood with varying degrees of substitution.

6. Figure 4e indicated an excitation wavelength dependent shift of the emission. Any

idea about which groups of the emission peaks are assigned, correspondingly?

A: Thanks for the comments. CX-Wood contains both the oxygen clusters of cellulose and the π -conjugated building blocks of the benzene ring in lignin. To further attribute the emission peaks of CX-Wood to specific components, we separately measured the RTP emission of CX-Lignin and CX-Cellulose under different excitation wavelengths. Spectral analysis reveals that emission peaks within 480-500 nm are predominantly attributed to CX-Cellulose, while features across 500-540 nm arise primarily from CX-Lignin. Critically, this excitation-dependent behavior of CX-Wood results from wavelength-selective activation of distinct chromophoric centers within each component.

We added this in the revised manuscript as **Supplementary Fig. 16**.

Supplementary Fig. 16. (a) RTP emissions of CX-Cellulose using different excitation wavelengths. (b) RTP emissions of CX-Lignin using different excitation wavelengths.

7. As concerning the 3D printing part, what are the challenges the authors have conquered, and based on what kind of strategy is not clear presented? Or just by mixing with water, and adjusting the concentration and viscosity? If it is by adjusting these parameters, the novelty of 3D printing of wood powders is not good enough.

A: Thanks for the comments.

Among the various AM techniques for wood, direct ink writing (DIW) has emerged as the most versatile 3D printing technique since it acts as a unique pathway to introduce

design freedom, multifunctionality, and stability simultaneously into the printed structures (*Nat Sustain.*, 2024, 7, 698-705; *Compos. B Eng.*, 2021, 225, 109249; *Prog. Mater. Sci.*, 2025, 152, 101462; *Adv. Mater.*, 2022, 34, 2108855; *Chem. Soc. Rev.*, 2023, 52, 1614-1649).

Generally, DIW requires the appropriate rheological behavior of the printed inks. Thus, wood is usually engineered to exhibit appropriate rheological behavior for printing by introducing thermoplastic plastics (such as PE, PP, PVC, PLA and so on) (*Compos. B Eng.*, 2020, 200, 108347; *Compos. B Eng.*, 2021, 219, 108873; *Adv. Compos. Hybrid Mater.*, 2023, 6, 81; *Adv. Drug Deliv. Rev.*, 107, 2016, 17-46). Additionally, the DIW inks can also be prepared from wood by extracting components (cellulose, hemicellulose and lignin) and subsequent modification for achieving appropriate rheological properties (*Carbohydr. Polym.*, 2020, 250, 116881; *Addit. Manuf.*, 2024, 92, 104397; *Ind. Crop. Prod.*, 2022, 186, 115234; *Int. J. Biol. Macromol.*, 2024, 267, 131364; *Sci. Adv.*, 2024, 10, eadk3250).

Nevertheless, using wood in the absence of external additives for DIW 3D printing was rarely reported. All these shortcomings motivated us to create a more convenient and sustainable 3D printable RTP formula using wood. Here, we simply modified the wood with carboxylic moieties for DIW printing, addressing the previous concerns. This is the novelty and progress.

We added this information to the revised manuscript.

8. There are papers have published about 3D printed carboxymethyl cellulose, such as *Carbohydrate Polymers*, 278, 2022, 118924 (doi.org/10.1016/j.carbpol.2021.118924.), and summarized in the reviewer paper (*Mater. Chem. Front.*, 2022, 6, 254-279). The authors need to clearly address the scientific advantage of the 3D print of the carboxymethylated wood.

A: Thanks for the comments. Firstly, we compared the differences in printing performance between CX-Wood and commercial carboxymethyl cellulose. Due to the presence of lignin in CX-Wood, which provides adhesive and structural support, the printed samples retain their shape, whereas structures printed with commercial

carboxymethyl cellulose are prone to collapse (*Adv. Mater.*, 2021, 33, 2001588) (**Supplementary Fig. 27**). Additionally, the tensile strength of commercial CMC is also significantly inferior to that of CX-Wood (**Supplementary Fig. 28**).

Secondly, we evaluated the phosphorescent properties of CX-Wood and commercial carboxymethyl cellulose. The commercial cellulose exhibited inferior phosphorescence intensity and phosphorescence lifetime compared to CX-Wood (**Supplementary Fig. 6**).

We added this to the revised manuscript as **Supplementary Fig. 27**, **Supplementary Fig. 28** and **Supplementary Fig. 6**.

Supplementary Fig. 27. Samples printed using commercial CMC and CX-Wood. As shown in the figure, the samples printed with CX-Wood retain their structural integrity, while those printed with commercial CMC exhibit varying degrees of collapse (highlighted in red).

Supplementary Fig. 28. Tensile strength of commercial CMC and CX-Wood.

Supplementary Fig. 6. (a) Afterglow emission spectra of commercial CMC and CX-Wood. (b) Lifetime decay profiles of commercial CMC and CX-Wood. Excitation wavelength = 340 nm.

9. According to the experimental section, the 3D printed wood powder is not post-cross-linked. Did the ink only contain wood and water? It is not clearly stated.

A: Thanks for the comments. CX-Wood ink is formulated only from chemically modified wood powder (CX-Wood) and deionized water. Upon hydration, hydrogen bond interactions were formed between -OH and -COOH in CX-Wood, determined by FT-IR spectra (**Supplementary Fig. 2**). The as-formed hydrogen bonds promoted the

association between these modified CX-wood powders, thereby establishing a stabilized network and merged interface without post cross-linking determined by SEM images (Supplementary Fig. 23).

We added this to the revised manuscript as Supplementary Fig. 2 and Supplementary Fig. 23.

Supplementary Fig. 2. FT-IR spectra of natural wood and different substituted degree CX-Wood.

Supplementary Fig. 23. SEM images of the samples printed using CX-Wood.

10. As the water evaporated after printing, will the structure shrink? If so, shrinking to

what degree? And how well can the morphology be reserved? Are any porous structures formed?

A: Thanks for the comments. The samples printed with CX-Wood have a certain structural shrinkage after drying, retaining ~90% of their original morphology (**Supplementary Fig. 24**). This was due to the collapse of the internal structure induced by the capillary force resulting from the evaporation of water during the drying process (*Carbon*, 2025, 232, 119826). The structure of the dried samples was evaluated using SEM, and a porous structure was observed inside the samples (**Supplementary Fig. 23**).

We added this to the revised manuscript as **Supplementary Fig. 24** and **Supplementary Fig. 23**.

Supplementary Fig. 24. Comparison of samples printed with CX-Wood before and after drying.

Supplementary Fig. 23. SEM images of the samples printed using CX-Wood.

11. How are the mechanical properties, such as elastic modulus, tensile stress, and compress stress of the 3D printed structures?

A: Thanks for the comments. We conducted tests on the tensile and compressive properties of CX-Wood. The tensile strength and Young's modulus of CX-Wood were 12.5 MPa and 151.0 MPa, respectively, while its compressive strength and compressive modulus were 8.7 MPa and 83 MPa (the pressure is 2000 N).

Additionally, we compared the mechanical strength between CX-Wood and natural woods (basswood and balsawood). Due to the anisotropic nature of natural woods (*Nat. Rev. Mater.*, 2020, 5, 642-666), they exhibit different tensile strengths in the "T-the radial direction of the wood ", "R-the tangential direction of the wood", and "L-the longitudinal direction of the wood". Natural basswood exhibits strong tensile performance only in the "L" direction, with tensile strength and Young's modulus of 70.6 MPa and 547.3 MPa, respectively. However, its tensile properties in both the T- and R-directions are inferior to those of CX-Wood. Natural balsa wood, in all three directions (T, R, and L), demonstrates lower tensile performance than CX-Wood.

Regarding compressive strength, both basswood and balsawood demonstrate strong compressive performance in the "TR" direction. Under a pressure of 2000 N, basswood exhibits compressive strength and compressive modulus of 8.4 MPa and 169 MPa respectively in the "TR" direction, while balsawood shows values of 6.1 MPa and 126.2

MPa in the same direction. However, the compressive properties of both woods in the "LR" direction are inferior to those of CX-Wood.

We added this to the revised manuscript as **Supplementary Fig. 25**.

Supplementary Fig. 25. (a) Schematic diagram of Tensile and compression testing directions for natural wood (basswood and balsawood). (b) Tensile (tensile strength and Young's modulus) properties of CX-Wood and basswood and balsawood. (c) Compressive (compress strength and compress modulus) properties of CX-Wood and basswood and balsawood (the pressure is 2000 N).

12. Is the natural wood grounded into powder? What is the size of the powder? Will the size affect the chemical modification degree?

A: Thanks for the comments. Natural wood powder with 60-mesh size was used in this article. To further understand the correlation between the size of the powder and chemical modification degree, a series of wood powder with different sizes were prepared. Generally, wood powders with reduced size exhibited higher substitution degree (Figure SX). This was because the reduced size was beneficial for its interaction with the liquid solution, thus, increasing the reaction efficiency.

We added this to the revised manuscript as **Supplementary Fig. 3**.

Supplementary Fig. 3. The DS of CX-Wood modified by wood powder particles of different mesh sizes.

13. If the modified RTP wood is in powder manner, how does the size affect the rheological behaviors of the wood ink?

A: Thanks for the comments. We evaluated the influence of the size of wood powder on the rheological behavior of CX-Wood. When the wood particle size is coarser than 20 mesh, no intersection occurs between G' and G'' , indicating that CX-Wood fails to transition from a solid to a liquid state (**Fig. 4b**). This occurs because coarser particles result in insufficient reaction due to their larger size. As the mesh number increased, both the viscosity and substitution degree of CX-Wood rose with decreasing wood powder particle size (**Fig. 4a**). Meanwhile, the crossover modulus ($G' = G''$) of CX-Wood decreased with increasing mesh number, indicating that CX-Wood ink is more likely to transform from a solid state to a liquid state at higher mesh counts. In summary, based on a comprehensive evaluation of wood particle size effects on the modification efficiency, RTP performance, and rheological behavior of CX-Wood, we selected 60-mesh basswood powder as the primary research object in this paper (**Supplementary Fig. 22**).

We added this to the revised manuscript as **Fig. 4a-b**, and **Supplementary Fig. 22**.

Fig. 4. (a) The dependence of viscosity of CX-Wood on shear rate for different sized particles. (b) Storage modulus (G') and loss modulus (G'') plotted versus shear stress of CX-Wood for different sized particles.

Supplementary Fig. 22. (a) The images of CX-Wood ink obtained after the modification of wood powder particles with different mesh sizes. (b) Phosphorescence spectra of CX-Wood modified by wood powder particles of different mesh sizes. (c) RTP decay profiles of CX-Wood modified by wood powder particles of different mesh sizes. Excitation wavelength = 340 nm.

14. Figure 2g, the mark of 80°C is confused, tested at the temperature, or the wood is heated at 80°C to remove the water? But this temperature is not supposed to remove all the water.

A: Thanks for the comments.

80 °C is not the temperature for the measure of RTP properties. The RTP properties were measured at room temperature.

We used the 80 °C to remove the water from the printed structure. ~1.43% water remained in the structure after the drying process.

To avoid such misunderstandings, we have changed "80 °C" to "Drying".

15. The discussion in 2.1 is suggested to be well-developed, currently, most of the writing is simply describing the phenomenon, instead of developed explanations. And many references are missing about all the xps, ftir peak assignments.

A: Thanks for the comments. For Figure 2a, we have cited References 56 (*Cellulose*, 1995, 2, 145-157) and References 57 (*Carbohydr. Polym*, 2021, 272, 118458) to validate the assignment of the -O-C=O functional group in the XPS spectra. For Supplementary Fig. 2, we cite references 58 (*Carbohydr. Polym*, 2020, 244, 116481) and References 59 (*Int. J. Biol. Macromol.*, 2021, 176, 72-77) to illustrate the enhancement of the carbonyl moieties in CX-Wood using FT-IR.

Reviewer #3 (Remarks to the Author):

Reviewer 3: The authors reported a concise strategy to construct room-temperature phosphorescence (RTP) woody materials by grafting carboxyl-functional groups onto native lignocellulosic matrices, endowing the RTP wood with the property of direct ink writing. The modified wood exhibited intense and long-lived RTP with a lifetime of 358.7 ms. The strategy is effective for the majority of woods. Impressively, a series of 3D RTP structures were flexibly designed and obtained using CX-Wood, and these obtained RTP structures could be recycled for 3D printing via water treatment. These RTP woods demonstrate a very promising source of structural materials for architecture, furniture and other applications. The results are of considerable interest, and potentially merit publication as a Nature Communication. However, there are a number of important points which must be addressed.

Thanks for the constructive comments.

1. As shown in Figure 1a, the modified wood is coated with sodium acetate ions, I wonder whether the ionic bonding also plays an important role in generating RTP?

A: Thanks for the comments. The ionic bonding does not affect the phosphorescence performance of CX-Wood. To verify this, we added additional Na^+ by adding NaCl into the CX-Wood system and determined the phosphorescence spectrum and phosphorescence lifetime. The results revealed no significant changes in the phosphorescent behavior of CX-Wood, confirming that ionic interactions do not influence the RTP properties. We added this to the revised manuscript as **Supplementary Fig. 9**.

Supplementary Fig. 9. (a) Phosphorescence spectra of CX-Wood and CX-Wood + Na⁺. (b) RTP decay profiles of CX-Wood and CX-Wood + Na⁺, excitation wavelength = 340 nm.

2. In Figure 2e, CX-Wood exhibited excitation-dependent RTP emission. Please explain the reason for this phenomenon.

A: Thanks for the comments. CX-Wood contains both the oxygen clusters of cellulose and the π -conjugated building blocks of the benzene ring in lignin. Thus, the excitation-dependent RTP emission should originate from the overlapping of lone-paired n-electron clouds in cellulose, the π - π^* transition of the benzene ring in lignin, and the conformational rigidity enhanced jointly by abundant intra- and intermolecular hydrogen bonds (*Chem. Soc. Rev.*, 2021, 50, 12616; *ACS Sustain. Chem. Eng.*, 2018, 6, 3169-3175).

To further validate this claim, we evaluated the RTP emission of both CX-Lignin and CX-Cellulose under different excitation wavelengths. It can be seen that short-wavelength excitation (250-310 nm) predominantly excites lignin chromophores, while long-wavelength excitation (310-430 nm) primarily activates cellulose chromophores. This wavelength-selective excitation process results in excitation-dependent emission characteristics of the material.

We added this to the revised manuscript as **Supplementary Fig. 16**.

Supplementary Fig. 16. (a) RTP emissions of CX-Cellulose using different excitation wavelengths. (b) RTP emissions of CX-Lignin using different excitation wavelengths.

3. The quantum yields of CX-Wood should be added.

A: Thanks for the comments. We have determined the quantum yields of CX-Wood with different degrees of substitution.

The phosphorescence quantum yield of the compound was obtained from the following equation:

$$\phi_{phos.} = \frac{B}{A} \times \phi_{PL}.$$

where A and B represent the integral areas of total photoluminescence and phosphorescence spectra, respectively. The phosphorescence was separated from PL by time-gating (*Nat. Mater.*, 2021, 20, 1539-1544).

The quantum yields of natural wood, CX-Wood with DS = 0.0664, CX-Wood with DS = 0.1009, CX-Wood with DS = 0.3288 and CX-Wood with DS = 0.5760 were measured to be 0.47%, 1.78%, 2.70%, 6.61%, and 8.32%, respectively. The phosphorescence quantum yield of natural wood, CX-Wood with DS = 0.0664, CX-Wood with DS = 0.1009, CX-Wood with DS = 0.3288 and CX-Wood with DS = 0.5760 were measured to be 0.25%, 0.93%, 1.55%, 3.86%, and 4.60%, respectively.

We added the result to the revised manuscript as **Supplementary Fig. 8.**

Supplementary Fig. 8. The quantum yields and phosphorescence quantum yields of natural wood and CX-Wood with varying degrees of substitution.

4. The mechanical strength of wood before and after the modification should be measured, such as compressive stress–strain curves.

A: Thanks for the comments. We conducted tests on the tensile and compressive properties of CX-Wood. The tensile strength and Young's modulus of CX-Wood were 12.5 MPa and 151.0 MPa, respectively, while its compressive strength and compressive modulus were 8.7 MPa and 83 MPa (the pressure is 2000 N).

Additionally, we compared the mechanical strength between CX-Wood and natural woods (basswood and balsawood). Due to the anisotropic nature of natural woods (*Nat. Rev. Mater.*, 2020, 5, 642-666), they exhibit different tensile strengths in the "T-the radial direction of the wood", "R-the tangential direction of the wood", and "L-the longitudinal direction of the wood". Natural basswood exhibits strong tensile performance only in the "L" direction, with tensile strength and Young's modulus of 70.6 MPa and 547.3 MPa, respectively. However, its tensile properties in both the T- and R-directions are inferior to those of CX-Wood. Natural balsa wood, in all three directions (T, R, and L), demonstrates lower tensile performance than CX-Wood.

Regarding compressive strength, both basswood and balsawood demonstrate strong compressive performance in the "TR" direction. Under a pressure of 2000 N, basswood

exhibits compressive strength and compressive modulus of 8.4 MPa and 169 MPa respectively in the "TR" direction, while balsawood shows values of 6.1 MPa and 126.2 MPa in the same direction. However, the compressive properties of both woods in the "LR" direction are inferior to those of CX-Wood.

We added this to the revised manuscript as **Supplementary Fig. 25**.

Supplementary Fig. 25. (a) Schematic diagram of Tensile and compression testing directions for natural wood (basswood and balsawood). (b) Tensile (tensile strength and Young's modulus) properties of CX-Wood and basswood and balsawood. (c) Compressive (compress strength and compress modulus) properties of CX-Wood and basswood and balsawood (the pressure is 2000 N).

5. For the preparation of different degree of substitution (DS) determination of CX-Wood, more experiment details should be given. Whether different degrees of substitution (DS) determination of CX-Wood depend on the reaction time or temperature?

A: Thanks for the comments. The degrees of substitution (DS) of CX-Wood depended on reaction time and reaction temperature.

We added the preparation method as **Table S1** and the results were added as

Supplementary Fig. 4.

Table S1. Preparation of CX-Wood with different Degrees of Substitution (DS).

Natural wood (g)	NaOH (g)	Alkali treatment time (min)	Alkali treatment temperature (°C)	ClCH ₂ COOH (g)	Ether-forming reaction time (min)	Ether-forming reaction temperature (°C)
Addition amount of NaOH						
5	1.5	60	35	7	90	80
5	3	60	35	7	90	80
5	4.5	60	35	7	90	80
5	6	60	35	7	90	80
Different alkali treatment time						
5	6	20	35	7	90	80
5	6	40	35	7	90	80
5	6	60	35	7	90	80
5	6	80	35	7	90	80
5	6	100	35	7	90	80
5	6	120	35	7	90	80
Different alkali treatment temperature						
5	6	60	15	7	90	80
5	6	60	25	7	90	80
5	6	60	35	7	90	80
5	6	60	45	7	90	80
5	6	60	55	7	90	80
5	6	60	65	7	90	80
Different ether-forming reaction time						
5	6	60	35	7	30	80
5	6	60	35	7	60	80
5	6	60	35	7	90	80
5	6	60	35	7	120	80
5	6	60	35	7	150	80
5	6	60	35	7	180	80
Different ether-forming reaction temperature						
5	6	60	35	7	90	40
5	6	60	35	7	90	50
5	6	60	35	7	90	60
5	6	60	35	7	90	70
5	6	60	35	7	90	80
5	6	60	35	7	90	90

Supplementary Fig. 4. Effect of reaction time and temperature on the DS of CX-Wood. (a) Alkali treatment time. (b) Alkali treatment temperature. (c) Ether-forming reaction time. (d) Ether-forming reaction temperature.

Response to the reviewers

Thank you very much for giving us this opportunity to revise our manuscript. We have fully addressed the concerns raised by the reviewers. The corresponding revisions were highlighted with yellow in the main text. Some figures were also revised. Please see also the point-by-point response to the reviewers' comments below. We hope now our manuscript can meet the requirements for publication in *Nature Communications*.

Reviewers' comments:

Reviewer #1 (Remarks to the Author):

I am not still convinced about the response on this manuscript. The response did not clearly discuss my concern regarding the novelty aspect and this is still not be considered as new concept. Rather, they have provided a lot of unwanted testing. Also, very recently the author published a similar article in Nature Communications (Nat Commun 16, 7978 (2025)). Again, there are a lot of works on wood and other biomass resources that have been focused on developing room temperature phosphorescent (RTP) (Nat. Commun. 14, 2614 (2023), Nat. Commun. 16, 868 (2025)). The author claimed that the novelty lied in the 3D printing part. However, 3d printing of wood has been reported (Science Advances 2024, 10, eadk3250). As I have seen in the paper from Thakur et al., they also did not use any additives. They printed wood from all-wood components. So, the author's claim of additive-free for the first time is not also correct. Considering the novelty and significance of the research, I do not recommend publishing this manuscript in Nature Communications.

A: Thanks for the comments. Thank you very much for once again reviewing our manuscript and providing valuable constructive feedback. We fully understand your stringent concerns regarding the novelty of our work and your deep appreciation of the field. We have thoroughly studied and reflected upon the outstanding work by Thakur et al. (Science Advances 2024, 10, eadk3250) and the numerous recent advances in wood-based Room Temperature Phosphorescence (RTP) that you rightly pointed out.

We acknowledge that achieving merely the 3D printing of wood or realizing RTP in wood individually has been reported before. However, it is difficult to obtain woody materials with RTP and printable properties simultaneously in a convenient manner.

Comparison between the materials from Thakur et al.'s Work and our work further demonstrated the result. The RTP lifetime from their materials was 28.97 ms. While, our materials showed the lifetime of 358.72 ms.

Actually, converting wood to RTP materials with flexibly designed structures is crucial but holds two challenges. One is how to tune the rheology properties of wood, so that it can be molded using DIW technology. The other is how to activate inherent RTP emission from wood powders.

Here, we found that chemical modification of wood can simultaneously enhance the RTP performance via increasing its spin-orbit coupling (SOC) value, determined by theoretical calculation (**Supplementary Fig.23**). Moreover, such modification can also enable the wood to be printable via DIW technology. Thus, we believe the core innovation and contribution of our work lie in "the integration of 3D printability with RTP using a fully biomass-based wood ink ."

Furthermore, we conducted a comparative life cycle assessment and cost analysis of CX-Wood against other reported materials for wood-based 3D printing. The results demonstrate the exceptional environmental performance of CX-wood ink, particularly its global warming potential (GWP) being only 27.6% of simulated wood ink and 56.4% of PALF ink. The GWP analysis confirms a significantly lower carbon footprint for CX-wood (12.03 kg CO₂ eq). Consequently, compared to currently reported wood-based 3D printing materials, CX-Wood ink production proves to be both cost-effective and environmentally sustainable. We added this to the revised manuscript as **Supplementary Fig. 5 and Table S2**.

We have revised the manuscript according to your comments to highlight our priorities. We sincerely thank you for helping us improve the quality of our paper.

Supplementary Fig. 23. Spin orbit coupling (SOC) values between the S₁ and T₁ states calculated for Lignin and CX-Lignin.

Supplementary Fig. 5. (a) Comparison of life cycle assessment results for the three inks; (b-d) Radar charts illustrating the contributions of different materials and energy inputs to the global warming potential (GWP) indicator (functional unit: production of 1.0 kg of ink).

Table S2. Phosphorescence lifetime vs. price: three "wood-based inks".

Types of Ink	Materials	RTP lifetime (ms)	Price (\$/kg)	Ref
CX-Wood inks	Balsawood; NaOH; Chloroacetic acid	358.72	22.5	This work
Simulated wood inks	Cellulose; nanocrystals; Cellulose; Nanofibers; Organosolv lignin	28.97	319	Science Advances 2024, 10, eadk3250
PALF inks	Lignocellulose; L-Lactic Acid; Choline chloride	72.74	53	Chemical Engineering Journal 2025, 523,168671

Reviewer #2 (Remarks to the Author):

Thanks to the authors for the efforts in addressing the concerns raised. However, several key issues remain insufficiently resolved.

1. One of the main questions is the novelty of the work as mentioned in Question 1. Unfortunately, the response appears to miss the core issue. Rather than addressing the novelty directly, the authors primarily compare CX-wood to CMC-based materials. The mention of CMC in the original question was intended to highlight that the chemical modification of wood to introduce carboxymethyl groups is a well-established and even commercialized technique. Therefore, the carboxymethylation of wood cannot be considered a novelty in this context.

A: We sincerely thank you for this insightful and constructive comment. You rightly pointed out the established nature of wood carboxymethylation, which has helped us to refine and better articulate the core novelty of our work. We fully agree that carboxymethylation, as a chemical modification technique for wood, is indeed well-established. However, the novelty of our work lies not in the carboxymethylation step.

Achieving easily printable and sustainable RTP materials from biomass sources, particularly wood, are crucial in the circle of RTP materials. Indeed, the conversion of wood into structurally customizable RTP materials faces two primary challenges: tuning the rheological properties of wood for molding via Direct Ink Writing (DIW), and activating the inherent RTP emission of wood powders. Here, we demonstrate that a specific chemical modification of wood addresses both challenges simultaneously. This treatment enhances RTP performance by increasing the spin-orbit coupling (SOC) value (**Supplementary Fig.23**), as determined by theoretical calculations, while also conferring excellent printability via DIW. Therefore, we believe the core innovation of our work lies in the integration of 3D printability with RTP using a fully biomass-based wood ink.

We have now revised the manuscript to unequivocally state this position. Thank you again for this essential feedback, which has helped us articulate the true significance of our work more clearly.

2. The second claimed novelty is the 3D printability of CX-wood without additional additives—simply by mixing with DI water. However, for facilitate this, what kind of scientific problem the authors have solved by developing what kind of new method? It seems the printability is facilitated by grafting the carboxymethyl groups. But this is nothing new.

A: We sincerely thank you for this insightful and constructive comment. You rightly highlighted that carboxymethylation is a well-established chemical modification, and we fully agree that merely applying it to wood and observing improved printability would, by itself, offer limited novelty.

Indeed, the ongoing shift from petroleum-based to bio-based manufacturing, driven by the need for sustainable development, is also reshaping the field of room temperature phosphorescence (RTP) materials. The development of easily printable and sustainable RTP materials from biomass resources, especially wood, has become an important academic goal. However, converting wood into structurally customizable RTP materials faces two major challenges: tuning the rheological properties of wood to enable molding via Direct Ink Writing (DIW), and activating the intrinsic RTP emission of wood powders.

In this study, we demonstrate that a specific chemical modification of wood simultaneously addresses both issues. This treatment not only enhances RTP performance by increasing the spin-orbit coupling (SOC) value (**Supplementary Fig.23**), as supported by theoretical calculations, but also imparts excellent printability via DIW. Therefore, we believe the core innovation of our work lies not in the use of carboxymethylation, but in revealing its unique dual-function role in wood, which unlocks a new platform for 3D-printable RTP wood materials.

Thank you once again for your valuable guidance, which has greatly helped us improve the scientific depth and clarity of our manuscript.

Supplementary Fig. 23. Spin orbit coupling (SOC) values between the S₁ and T₁ states calculated for Lignin and CX-Lignin.

3. Moreover, for the view of printable phosphorescence wood or cellulose materials, current system also does not present an obviously advancement. The manuscript introduces limited new insights or strategies to achieve RTP, as quite many previous work reported in this field, to name a few Nat. Commun. 14, 2614 (2023), Nat. Commun. 16, 868 (2025), Adv. Mater. 2023, 35, 2304032, Adv. Funct. Mater. 2024, 34, 2403977.

Given these, the revised manuscript does not demonstrate a clear improvement in terms of novel findings or methodological innovations that effectively address a specific scientific gap in the field.

Based on this, the selling points of current work is not strong. And the novelty declared here is not convincing. Therefore, it is not suggested to be published in Nature Communications.

A: Thank you for your comments.

In most reported studies, RTP emission from wood has been achieved primarily through rigidification strategies. These methods enhance the rigidity of the molecular environment, thereby suppressing non-radiative decay of triplet excitons and improving the RTP performance of wood-derived components.

In contrast, during our exploration, we uncovered a distinct phenomenon: the spin-orbit coupling (SOC) value of lignin can be significantly enhanced by introducing carboxylic functional groups. This finding offers a new avenue for the future molecular design of wood-based RTP materials. Moreover, the incorporation of carboxylic moieties into wood powders also enables effective tuning of rheological properties, making the material suitable for Direct Ink Writing (DIW).

From this perspective, our work presents a new solution: a material platform that inherently integrates both functionality (RTP) and processability (DIW). We believe this "all-in-one" paradigm, directly transforming a raw natural resource into a designable phosphorescent ink, represents a novel finding with substantial implications for the future of sustainable room-temperature phosphorescent materials and biomass-based additive manufacturing.

We have revised the manuscript to clarify this conceptual distinction and underscore our unique contribution. We are confident that this work marks a meaningful and instructive advancement in the field of sustainable functional materials.

And few more concerns as following:

4. The discussion and presented data in the manuscript is missing a focal point. Although the authors declare the point of current work is the 3D printing of CX-wood. However, more than two-thirds of the results and discussion of the manuscript is devoted to the phosphorescence, and two out of four main figures exclusively present phosphorescence data. The remaining one-third discussion covers multiple properties of the printed structure, such as mechanical properties, flame-retardant properties, decomposition etc. The layout and data presentation result in in a scattered manuscript lacking a clearly defined focal point.

A: Thank you for the comments. Accordingly, we have moved the sections concerning mechanical properties, flame-retardant properties, and related content to the Supplementary Information (SI). This adjustment allows the manuscript to focus more effectively on its core novelty: the simultaneous integration of RTP functionality and printability into a single wood-derived material.

5. The 3D print ink is composed of 60 mesh-size CX-wood micro-particle dispersed in DI water. The ink is not a polymer solution, but a particle-based suspension. This kind of fluid is usually a dilatant liquid, exhibiting shear-thickening (dilatant) behavior, in which shear stress increases with higher shear rates. However, the rheological property characterized here show the property of pseudoplastic fluid. The authors are suggested to explain and dig deep about this.

A: Thank you for your comments. Indeed, the dispersion of CX-Wood in aqueous solution should not be regarded as a simple particle system. Natural wood is composed of three primary components, cellulose, hemicellulose, and lignin. During the carboxylic modification process, the acid or base treatment partially disrupts the original supramolecular structures formed by these components, yielding either grafted polymer chains (resembling a “brush-like” structure) or free polymer chains. (*Nanoscale*, 2011, 3, 71-85) These resulting structures engage in electrostatic and intermolecular interactions, which contribute to the high viscosity of the system. When shear stress is applied, these interactions are weakened, leading to a progressive decline in viscosity as shear rate increases. Thus, this shear-thinning behavior of CX-Wood is characteristic of pseudoplastic fluids.

We have revised the manuscript to clarify this.

6. The authors compared the mechanical properties of the printed structures based on CX-wood and CMC. The mechanical property is depended on the porous structure. It is not reasonable to compare this without talking about the internal morphology of the printed structures.

A: Thanks for the comments. We tested the internal morphology of the structure printed with commercial CMC. Compared to the internal structure of CX-Wood, the commercial CMC exhibits an uneven structure, relatively thin pore walls, and partial areas with fractures. Structural defects are present, therefore resulting in inferior mechanical properties.

We added this to the revised manuscript as **Supplementary Fig. 31**.

Supplementary Fig. 31. SEM images of the samples printed using commercial CMC.

7. Similarly, for the flame-retardant property, which is typical feature for materials with porous structure as reported in many aerogel materials, for instance, *Chemical Engineering Journal* 499 (2024) 155939. The flame-retardant effect observed here may stem from porosity or other physical factors rather than only intrinsic material properties. The authors are encouraged to explore and clarify the underlying mechanism.

A: Thanks for the comments. The flame retardancy of the CX-Wood printing material stems from the dual effects of its "porous structure" and "post-combustion char layer":

The abundant internal mesoporous structure exhibits exceptional thermal insulation properties, effectively blocking the transfer of combustion heat to the material's interior and preventing deep-seated combustion. Simultaneously, the small pore size suppresses gas flow, reducing oxygen penetration into the combustion zone and directly weakening the combustion intensity. Furthermore, during combustion, the material forms a dense and stable char layer that continuously impedes the transfer of heat and oxygen to prevent further burning. This char layer allows the material to maintain its basic structure after combustion, avoiding severe collapse or melting (*Chem. Eng. J.*, 2024, 499, 155939).

These two mechanisms work synergistically to enhance the material's overall flame retardant performance. We added it into the revised manuscript.

Reviewer #3 (Remarks to the Author):

I agree to the revised version.

A: Thanks for the comments.

Response to the reviewers

Dear editor,

Thank you once again for the opportunity to submit a revised version of our manuscript. We have carefully addressed the concerns raised by the reviewers. The corresponding revisions were highlighted with yellow in the main text. Some figures were also revised. Please see also the point-by-point response to the reviewers' comments below. We hope now our manuscript can meet the requirements for publication in *Nature Communications*.

Reviewers' comments:

Reviewer #2 (Remarks to the Author):

In the revised version, the authors have not made substantial changes to the manuscript either about the novelty or the results. Instead, they mainly added further explanations/augments about the claimed novelty. In the revised version, there is no strong evidence or data to significantly enhance the scientific importance of the manuscript. The newly added data are a life cycle assessment (LCA) methodology and a DFT simulated spin orbit coupling (SOC) values, both presented in the supporting information.

The former one could be served as an add-on, industry-oriented consideration, but it does not strengthen the scientific novelty of the study. Regarding the DFT results, the phosphorescent emission inherently involves intersystem crossing from the singlet to the triplet state, a process that is universally governed by SOC, which is not specific to the present system. Moreover, the enhancement of lifetime may originate from multiple factors. It is a risk to ascribe the reason to the enhanced SOC value, and the value is obtained by simulation.

Moreover, as mentioned in the previously round of peer-review, both the 3D printing of wood and the realization of room temperature phosphorescence (RTP) in wood have been reported individually in previously publications. In revised work, the authors declare the novelty based on combining these two aspects to fabricate 3D-printed wood-

based materials exhibiting RTP. To enable this functionality, the authors employ carboxymethylation of wood. However, this is a well-established and widely used chemical modification method for wood-based materials. There is not concept or strategy developed here. On the other hand, as concerning the RTP performance, the lifetime is on the order of hundreds of ms. In the field of organic RTP materials, this performance is relatively ordinary. The reported lifetime has been in the range of seconds even in the publications reported couple of years ago.

Thus, the novelty, the development of a new scientific concept or methodological strategy, and the RTP performance of the revised manuscript are not sufficiently convincing. It is not suggested to be published in Nature Communications.

A: Thank you for the comments.

Regarding the comment that **“the enhanced lifetime may originate from multiple factors. It is risky to ascribe the reason to the enhanced SOC value, and the value is obtained by simulation”** we agree that a multi-faceted explanation is more robust.

To further substantiate the discussion, we have now performed additional calculations of the singlet-triplet energy gap (ΔE_{ST}) for CX-Wood with varying degrees of substitution. The results show a clear decrease in ΔE_{ST} with increasing substitution, which provides an additional, thermodynamically favorable factor for enhanced intersystem crossing (*Adv. Funct. Mater.* 2024, 34, 2407420).

Thus, combined with the FT-IR data (**Supplementary Fig. 2**), DFT-calculated binding energy (**Fig. 3c**), and transient absorption spectra (**Supplementary Fig. 22 and Supplementary Fig. 23**), these results clearly demonstrate that the enhanced RTP performance originated from strengthened internal interactions, promoted spin-orbit coupling, reduced ΔE_{ST} , and increased intersystem crossing (ISC) efficiency.

Table S4 The singlet-triplet energy gap (ΔE_{ST}) of Natural wood and CX-Wood.

	DS	$\lambda_{fluo.}$ (nm)	$^a)E_{S1}$ (eV)	$\lambda_{phos.}$ (nm)	$^b)E_{T1}$ (eV)	$^c)\Delta E_{ST}$ (eV)
Natural wood	0	410	3.024	515	2.408	0.616
	0.0664	445	2.787	513	2.417	0.370
CX-Wood	0.1009	450	2.756	510	2.431	0.325
	0.3288	455	2.725	508	2.441	0.284
	0.5760	460	2.696	505	2.455	0.241

a) $E_{S1} = 1240/\lambda_{fluo.}$; b) $E_{T1} = 1240/\lambda_{phos.}$; c) $\Delta E_{ST} = E_{S1} - E_{T1}$ ¹². (*Nat. Commun.*, 2025, 16, 7978)

Regarding the comment that “as concerning the RTP performance, the lifetime is on the order of hundredsof ms. In the field of organic RTP materials, this performance is relatively ordinary”, we agree that the lifetime of several hundred milliseconds, while functionally significant, does not set a record within the broad field of organic RTP materials where crystalline or heavily doped systems can achieve seconds-long afterglow.

However, the significance of our work lies in a distinct and targeted material category. First, the RTP emission of CX-Wood originates intrinsically from its lignin component without incorporating any external phosphorescent chromophores. Within this specific class of intrinsic, fully bio-based RTP materials, the achieved lifetime is competitive. Second, and more critically, when compared to state-of-the-art 3D-printable wood-based or cellulose-derived materials, CX-Wood demonstrates a clear and notable advantage in phosphorescent lifetime. This highlights the effectiveness of our strategy in unlocking and enhancing the innate photophysical properties of wood within a scalable manufacturing platform (**Table S2**).

Therefore, although the lifetime of CX-Wood may not rival that of traditional, highly optimized organic RTP material systems, it demonstrates clear application potential and practical significance within the emerging field of developing intrinsically phosphorescent, sustainable, and fully 3D-printable bulk biomaterials.

We added this to the revised manuscript as **Table S4**.

Reviewers' comments:

Reviewer #3 (Remarks to the Author):

Since single-crystal structures cannot be obtained for this material system, specific intermolecular interactions and molecular packing arrangements remain undetermined. The theoretical calculations presented in the article are all based on computational simulations, and their confirmatory value is open to debate. Only by performing simulations on a large number of molecules can the results approach closer to the real situation. Therefore, the accuracy of the DFT (Density Functional Theory) calculations is subject to further discussion.

Phosphorescent wood reported typically involved creating models through 3D printing and then soaking them in phosphorescent dyes. This work proposes directly printing modified wood into 3D models, ensuring that both the interior and exterior of the models consist of phosphorescent material. I believe this work has significant potential to advance the industrialization of phosphorescent materials. I agree to the revised version.

A: We thank the reviewer for pointing out that analyzing intermolecular interactions based on small molecular models may not sufficiently capture the true interaction features of the system. In response to this important comment, we have systematically supplemented and improved the computational models and analysis methods.

In the revised manuscript, instead of using isolated molecular pair models, we constructed a periodic crystal model containing 20 molecules and introduced periodic boundary conditions to simulate an infinitely extended bulk structure. This approach allows for the simultaneous consideration of long-range effects and many-body cooperative interactions. Based on this periodic system, we calculated the total interaction energy among the 20 molecules within the unit cell, yielding results that are statistically and physically more reliable than those from small molecular cluster models. Furthermore, we performed additional analysis using the Independent Gradient Model (IGM) method on the periodic system to visualize the spatial distribution and characteristics of the intermolecular interactions.

The results of these large-scale simulations are consistent with our initial

calculations on smaller models. This confirms the robustness of our original conclusions regarding the enhanced intermolecular interactions in CX-Wood and their correlation with the photophysical properties.

We added this to the revised manuscript as **Supplementary Fig. 21**.

Supplementary Fig. 21. Calculations based on the periodic model containing 20 molecules. (a) Calculated interaction and (b) independent gradient models of Cellulose—Cellulose (Cell—Cell) and CX-Cellulose—CX-Cellulose (CX-Cell—CX-Cell); (c) Calculated interaction and (d) independent gradient models of Lignin—Lignin (Lig—Lig) and CX-Lignin—CX-Lignin (CX-Lig—CX-Lig). (e) Calculated interaction and (f) independent gradient models of Cellulose—Lignin (Cell—Lig) and CX-Cellulose—CX-Lignin (CX-Cell—CX-Lig).

Response to the reviewers

Reviewers' comments:

Reviewer #3 (Remarks to the Author):

The revisions have resolved my issues, and I agree to publish the work.

Answer: We thank the reviewer for these comments.